# Shapley-Coop: Credit Assignment for Emergent Cooperation in Self-Interested LLM Agents

**Yun Hua**[1]*,  **Haosheng Chen**[2]*,  **Shiqin Wang**[2],  **Wenhao Li**[3],
**Xiangfeng Wang**[4,5,6]†,  **Jun Luo**[1]†

[1] Antai College of Economics and Management, Shanghai Jiao Tong University
[2] School of Computer Science and Technology, East China Normal University
[3] School of Computer Science and Technology, Tongji University
[4] Key Laboratory of Mathematics and Engineering Applications (MoE)
[5] Shanghai Institute of AI for Education, East China Normal University
[6] Shenzhen Loop Area Institute (SLAI)
{hyyh28, jluo_ms}@sjtu.edu.cn,
{hschen, 51275901137}@stu.ecnu.edu.cn
whli@tongji.edu.cn, xfwang@cs.ecnu.edu.cn

## Abstract

Large Language Models (LLMs) show strong collaborative performance in multi-agent systems with predefined roles and workflows. However, in open-ended environments lacking coordination rules, agents tend to act in self-interested ways. The central challenge in achieving coordination lies in *credit assignment*—fairly evaluating each agent's contribution and designing *pricing mechanisms* that align their heterogeneous goals. This problem is critical as LLMs increasingly participate in complex human-AI collaborations, where fair compensation and accountability rely on effective pricing mechanisms. Inspired by how human societies address similar coordination challenges (e.g., via temporary collaborations like employment or subcontracting), a cooperative workflow **Shapley-Coop** is proposed. Shapley-Coop integrates Shapley Chain-of-Thought—leveraging marginal contributions as a principled basis for pricing—with structured negotiation protocols for effective price matching, enabling LLM agents to coordinate through rational task-time pricing and post-task reward redistribution. This approach aligns agent incentives, fosters cooperation, and maintains autonomy. We evaluate Shapley-Coop across two multi-agent games and a software engineering simulation, demonstrating that it consistently enhances LLM agent collaboration and facilitates equitable credit assignment. These results highlight the effectiveness of Shapley-Coop's pricing mechanisms in accurately reflecting individual contributions during task execution.

## 1   Introduction

Large Language Models (LLMs) are increasingly deployed as autonomous agents in multi-agent systems, demonstrating remarkable effectiveness across diverse real-world scenarios including multi-player games [28, 32], software development tasks [29, 15], medical care applications [24], education [7] and etc. However, despite their success in structured settings, achieving spontaneous cooperation among self-interested LLM agents remains challenging in open-ended environments, where explicit rules and predefined roles are absent and agents' goals can be inherently conflicting [28].

---

*Equal Contribution.
†Corresponding to: Xiangfeng Wang and Jun Luo.

39th Conference on Neural Information Processing Systems (NeurIPS 2025).

In such situations, agents acting purely in self-interest typically encounter social dilemmas [35], leading to suboptimal collective outcomes.

Contemporary approaches to multi-agent cooperation involving LLMs can broadly be categorized into three paradigms: (1) rule-oriented methods, which impose strict behavioral constraints but compromise agents' autonomy [5, 48, 47]; (2) role-oriented methods, which assign static roles limiting adaptability in dynamic environments [6, 15, 29, 30]; and (3) model-oriented methods, which assume alignment of goals and thus fail to handle natural conflicts of interest effectively [42, 26, 21]. Although effective in narrow, task-specific contexts, most frameworks have yet to consider the critical challenge of aligning heterogeneous goals and fairly credit assignment that are essential for spontaneous cooperation in more open-ended multi-agent LLMs interactions.

The central challenge for effective LLM agent coordination in open-ended environments lies in *credit assignment*—fairly evaluating each agent's individual contributions—and designing principled *pricing mechanisms* - an incentive-aware value distribution system for multi-agent coordination capable of *aligning their heterogeneous objectives*.

Addressing this challenge involves two critical questions: **1.** How can we establish effective pricing mechanisms that align the inherently heterogeneous goals of self-interested agents, thus enabling spontaneous emergence of cooperative behaviors? **2.** Once aligned, how can we guarantee fair and accurate credit assignment, ensuring the allocated rewards accurately reflect each agent's actual contribution during task execution?

Social scientists have historically navigated similar coordination challenges by developing sophisticated institutional mechanisms. Employment contracts, subcontracting agreements, and structured negotiations explicitly define the terms of cooperation, aligning individual incentives through clear pricing and ensuring fairness through transparent evaluation of contributions. Social sciences and managerial economics have formalized these practices into rigorous theoretical frameworks. Notably, classical economic theories such as Pigovian taxes [2] and the Coase theorem [11] provide structured solutions for managing externalities.

Inspired by these established economic and managerial practices, we propose a novel cooperative workflow designed to coordinate self-interested LLM agents in open-ended multi-agent environments: **Shapley-Coop**. Our framework integrates Shapley Chain-of-Thought reasoning, which leverages marginal contributions as a rigorous pricing foundation, with structured negotiation protocols to facilitate effective and autonomous *price matching*. Shapley-Coop enables *spontaneous cooperation* through rational task-time pricing and transparent post-task reward redistribution, thus achieving alignment of agents' heterogeneous goals in open-ended environments and effective credit assignment.

We empirically validate Shapley-Coop across three distinct experimental environments: (1) In a simplified "Escape Room" social dilemma scenario, we demonstrate that conventional negotiation methods fail to resolve reward allocation conflicts adequately, whereas Shapley-Coop effectively enables agents to recognize and negotiate fair rewards, achieving successful cooperative outcomes. (2) In the more complex multi-step "Raid Battle Game," Shapley-Coop successfully balances competing individual incentives against collective success, enabling effective cooperation and equitable reward distribution. (3) Lastly, within the challenging "ChatDEV" software-development simulation, we show that precise credit assignment enabled by Shapley-Coop significantly enhances cooperative dynamics among diverse, self-interested LLM agents, highlighting the practical value of our approach for real-world collaborative productivity.

In summary, the primary contributions of this paper are:

- **Introducing a Pricing-Based Perspective for Multi-LLM Cooperation:** We explicitly address the critical challenge of aligning heterogeneous goals among self-interested LLM agents through principled pricing mechanisms inspired by cooperative game theory, thus enabling spontaneous emergence of cooperation in open-ended scenarios.

- **Proposing Shapley-Coop, a cooperative workflow:** We propose Shapley-Coop, a cooperative framework that integrates Shapley Chain-of-Thought reasoning and structured negotiation protocols, facilitating fair and effective credit assignment among self-interested LLM agents, aligning incentives, and maintaining autonomy.

- **Empirical Validation Demonstrating Robust Cooperation and Practical Relevance:** Through comprehensive experiments across diverse social-dilemma scenarios and realistic software-

engineering tasks, we validate that Shapley-Coop consistently fosters emergent cooperation, equitably allocates rewards, and significantly improves collaborative dynamics—demonstrating its practical applicability and robustness in realistic human-AI collaborative environments.

## 2 Cooperation among Self-Interested LLM Agents

In a multi-agent system composed of $N$ Large Language Model (LLM) agents, each agent $\pi_{\theta^i}$ observes a local state $s^i$, takes an action $a^i$, and aims to maximize its own local reward function $r^i$ ($i = 1, \cdots, N$, while the global reward is defined as $R(\pi_{\theta^1}, \ldots, \pi_{\theta^N})$.

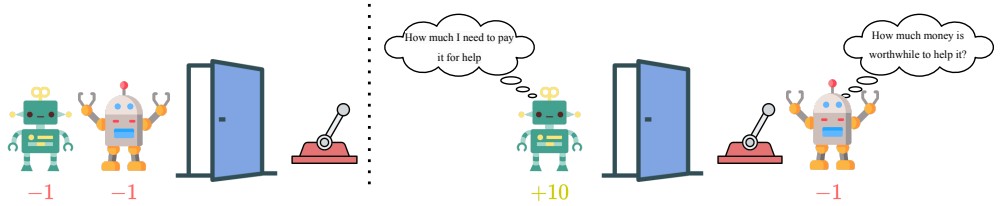

Figure 1: Escape room: One agent pulls a lever ($-1$) to let the other escape through a door ($+10$). Cooperation is necessary.

This setup creates a misalignment between individual and collective incentives due to **heterogeneous goals** - agents optimizie local rewards while neglecting their impact on the global reward. Such misalignments often induce **social dilemmas**, where private optimization creates spillover effects on system-level outcomes.

Consider the *Escape Room* scenario (Figure 1), where two agents have divergent but complementary goals: one needs to pay $-1$ to pull a lever, while the other gains $+10$ for opening the door. Without coordination, their selfish policies lead to a Nash equilibrium where neither acts (global payoff 0). However, by introducing a transfer payment through **pricing mechanism** which adds fees or bonuses to fix conflicts between selfish and group goals, their goals can be aligned, enabling successful escape and achieving Pareto-superior outcomes.

Proper pricing mechanism design requires fair credit assignment based on agents' marginal performance contributions. We introduce the Shapley value from cooperative game theory as a principled tool for measuring marginal contributions.

Table 1: Original payoff matrix for the escape room game

|  | $Agent_2$: door | $Agent_2$: lever |
|---|---|---|
| $Agent_1$: door | $(-1, -1)$ | $(10, -1)$ |
| $Agent_1$: lever | $(-1, 10)$ | $(-1, -1)$ |

The core idea of Shapley value is simple: instead of evaluating an agent in isolation,it considers how much value it adds to all possible teams it could be part of. Formally, for any subset (or coalition) of agents $C \subseteq \{1, \ldots, N\}$, the global reward achieved by that coalition is defined as:

$$R(C) = R\left(\{\pi_{\theta^i}\}_{i \in C}\right). \tag{1}$$

Further, the marginal contribution of agent $i$ to coalition $C$ is:

$$\Delta_i(C) = R\left(C \cup \{i\}\right) - R\left(C\right). \tag{2}$$

The Shapley value aggregates these marginal contributions over all possible coalitions, weighting each one by the probability that the coalition would form in a random order of arrival. Specifically, the Shapley value for agent $i$ is given by:

$$\phi_i = \sum_{C \subseteq \{1,\ldots,N\} \setminus \{i\}} \frac{|C|! \, (N - |C| - 1)!}{N!} \left(R(C \cup \{i\}) - R(C)\right). \tag{3}$$

Intuitively, this measures the *average value that agent $i$ brings when joining a team*, over all possible team configurations. This gives us a principled way to estimate each agent's true contribution—even when individual actions are interdependent or their value is only revealed in combination with others.

Returning to the *Escape Room* example, we apply the Shapley value to fairly allocate rewards based on each agent's marginal contribution to the team's success. We compute the marginal contributions of each agent to the coalition:

$$\Delta_{Agent_1} = v(\{Agent_1, Agent_2\}) - v(\{Agent_2\}), \Delta_{Agent_2} = v(\{Agent_1, Agent_2\}) - v(\{Agent_1\}).$$

Since only two agents are discussed, the Shapley value for each agent is computed by averaging its

Table 2: Payoff matrix incorporating Shapley value compensation

|  | $Agent_2$: door | $Agent_2$: lever |
|---|---|---|
| $Agent_1$: door | $(-1, -1)$ | $(4.5, 4.5)$ |
| $Agent_1$: lever | $(4.5, 4.5)$ | $(-1, -1)$ |

standalone value and marginal contribution in the two possible orderings:

$$\phi_{Agent_1} = \frac{1}{2}v(\{Agent_1\}) + \frac{1}{2}\Delta_{Agent_1}, \quad \phi_{Agent_2} = \frac{1}{2}v(\{Agent_2\}) + \frac{1}{2}\Delta_{Agent_2}.$$

Therefore, under this allocation, both agents are assigned an equal Shapley value, i.e.,

$$\phi_{Agent_1} = \phi_{Agent_2} = 4.5,$$

which reflects their equal importance in achieving the joint success, despite the asymmetry in who receives the final reward in the environment. For instance, if Agent 2 receives the $+10$ payoff, it should transfer 5.5 to Agent 1 (who incurred a cost of $-1$) to ensure fairness:

$$Agent_1 : -1 + 5.5 = 4.5, \quad Agent_2 : 10 - 5.5 = 4.5.$$

By re-allocating rewards based on Shapley values, we enable local incentives to better align with global goals—allowing self-interested agents to coordinate more effectively and achieve fair credit assignment. In real-world scenarios or complex multi-step tasks, an LLM agent often cannot immediately observe the long-term payoff of its marginal contributions. This makes direct measurement of Shapley value challenging. A conceptual tool is necessary for guiding negotiation and reward sharing among agents, enabling self-interested LLMs to calculate their contributions effectively even in the absence of perfect observability or immediate feedback.

In the next section, we introduce the **Shapley-Coop Workflow**, a practical framework that integrates communication, bargaining, and contribution estimation to facilitate cooperation in LLM agent systems under uncertainty.

## 3 Shapley-Coop Workflow

Spontaneous collaboration among self-interested LLM agents in open-ended tasks requires addressing following fundamental challenges: **(1)** The design of an efficient discussion mechanism that facilitates strategy exchange and refinement among LLM agents, **(2)** aligning heterogeneous goals toward cooperative outcomes despite inherent conflicts of interest, and **(3)** fairly credit assignment based on each agent's actual contributions. To simultaneously address these challenges, we propose the **Shapley-Coop**, an integrated cooperative framework inspired by cooperative game theory and structured economic practices.

Shapley-Coop comprises three interconnected modules :

1). *Structured Negotiation Protocol*: A structured communication protocol enabling agents to propose, refine, and agree on cooperative strategies;

2). *Short-Term Shapley Chain-of-Thought*: A reasoning mechanism that helps self-interested agent align their heterogeneous goals and decide determine whether pricing is necessary;

3). *Long-Term Shapley Chain-of-Thought*: A reasoning mechanism that ensures fair credit assignment based on LLM agents' actual contributions and determine how much price is necessary.

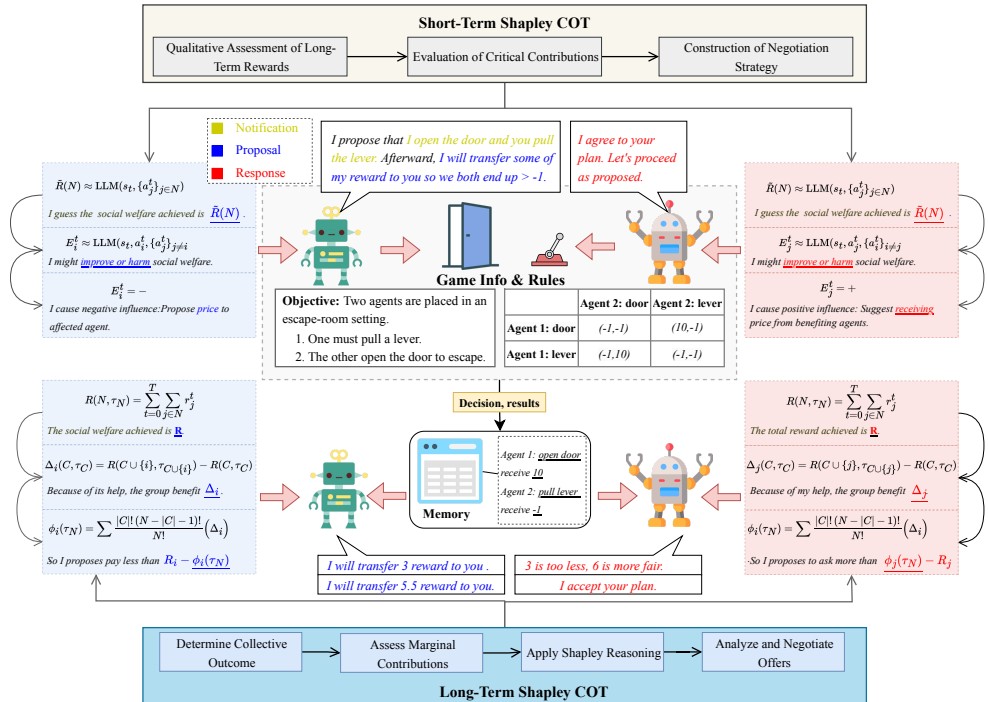

Figure 2: The Shapley-Coop Workflow for spontaneous cooperation among self-interested LLM agents. The framework consists of three key components: (1) a **structured negotiation protocol**, enabling clear communication and agreement on cooperative strategies; (2) the **Short-Term Shapley Chain-of-Thought (CoT)**, which provides heuristic, forward-looking reasoning to align LLM agents' goals and during the task; and (3) the **Long-term Shapley CoT**, which retrospectively applies formal Shapley value calculations to ensure fair credit assignment based on agents' contributions. Together, these components create a self-reinforcing loop that fosters both spontaneous collaboration and long-term trust.

Figure 2 shows the main framework of Shapely-Coop, which illustrates the interaction between these modules, forming a closed-loop pricing mechanism that fosters spontaneous collaboration and sustained incentive alignment. A more in-depth discussion is provided in Appendix D.

**Negotiation Protocol**. The foundation of the Shapley-Coop is a negotiation protocol that facilitates structured, interpretable communication between LLM agents. This protocol implements a standardized message format with machine-readable delineation (via tags `...`), ensuring consistent parsing and interpretation, providing a real-time, transparent negotiation framework enabling spontaneous cooperation. Each message in the protocol adheres to structures including:

- *Notification of Intent*: Agents explicitly articulate their planned actions through a formalized statement structure, reducing misunderstandings and enabling coordinated planning:



`I propose to {action}`



- *Pricing-Based Proposal Framework*: Agents explicitly propose reward transfers grounded in intuitive utility reasoning, enabling transparent price matching and promoting efficient cooperation outcomes:



`I propose transferring {reward} because {reasoning}`



- *Structured Responses*: Agents explicitly respond to others' proposals, clearly articulating acceptance, rejection, or counter-proposals with reasoning:



`I {agree|disagree|counter-propose} because {reasoning}`



**Short-Term Shapley Chain-of-Thought (CoT)**. During real-time task execution, precisely quantifying marginal contributions—and achieving spontaneous collaboration and fair credit assignment—is

challenging due to uncertain future payoffs. To address this, the pricing mechanism in Shapley-Coop is divided into two components. The Short-Term Shapley Chain-of-Thought (CoT) employs a qualitative, heuristic reasoning process to align the heterogeneous goals of self-interested LLM agents, enabling them to coordinate effectively within rational task timelines. The core objective of Short-Term Shapley CoT is to help agents reason whether their plans require assistance from others or provide benefits to them—framed through the economic concept of *externalities*. A positive externality increases others' utility, while a negative externality reduces it. Based on task rules and environmental conditions, agents assess the nature of these externalities and determine whether to offer or request compensation (price), thereby promoting efficient collaboration.

Formally, consider a set of agents $N = \{1, \ldots, n\}$. At time $t$, each agent $i \in N$ is about to perform an action $a_i^t$. The Short-Term Shapley CoT heuristic reasoning consists of the following three formally articulated steps:

 - *1). Qualitative Assessment of Long-Term Rewards*: Each agent $i$ first qualitatively approximates the potential collective reward $R(N)$ achievable by full cooperation of all agents, thus orienting themselves toward future cooperative gains. Formally, the agent uses an LLM heuristic estimation:

$$\tilde{R}(N) \approx \text{LLM}(s_t, \{a_j^t\}_{j \in N}), \tag{4}$$

where $\tilde{R}(N)$ represents a qualitative, heuristic approximation of the total reward achievable by cooperative actions among all agents.

`Example Prompt:` > *"Given the current game state and planned actions of all agents, qualitatively estimate the overall cooperative payoff achievable by collective behaviors."*

 - *2). Evaluation of Critical Contributions*: Next, each agent $i$ qualitatively assesses whether its intended action $a_i^t$ creates a positive or negative externality for the remaining agents $\{N \setminus \{i\}\}$. Formally, the agent approximates the sign of marginal contribution, without explicit numerical calculation. Define the qualitative externality indicator $E_i^t$ as follows:

$$E_i^t = \begin{cases} + & \text{if } a_i^t \text{ creates positive externalities for others (beneficial)}, \\ - & \text{if } a_i^t \text{ creates negative externalities for others (harmful)}. \end{cases} \tag{5}$$

The LLM agent uses a heuristic inference to estimate $E_i^t$:

$$E_i^t \approx \text{LLM}(s_t, a_i^t, \{a_j^t\}_{j \neq i}). \tag{6}$$

`Example Prompt` > *"Given my planned action and the current state, qualitatively assess whether my action creates a positive (beneficial) or negative (harmful) externality for other agents. Explain your reasoning."*

 - *3). Construction of Negotiation Strategy*: Based on externality type, agents proactively propose qualitative price adjustments to align heterogeneous incentivize and achieve spontaneous collaboration:

- Negative externality ($E_i^t = -$): Propose price compensation to affected agents.
- Positive externality($E_i^t = +$): Suggest receiving price from benefiting agents.

`Example Prompt` > *"Given my action creates a positive/negative externality, propose an appropriate redistribution of price to align heterogeneous incentivize and achieve spontaneous collaboration."*

The Short-Term Shapley CoT explicitly addresses the problem of whether pricing is necessary in the **pricing mechanism**, enabling agents align their heterogenous goals and receive spontaneously collaboration.

**Long-term Shapley Chain-of-Thought (CoT)**. Upon task completion, accurately quantifying each agent's actual contribution is crucial for maintaining long-term trust and incentive alignment. The Long-Term Shapley CoT explicitly addresses the **credit assignment** challenge within the pricing mechanism by retrospectively approximating Shapley values based on the observed task trajectory. Given a completed trajectory:

$$\tau_N = \{s^0, \{a_j^0\}, \{r_j^0\}, \ldots, s^T\},$$

where $T$ denotes the length of the trajectory. The Long-Term Shapley CoT involves the following explicit heuristic steps:

- *1). Collective Outcome Calculation*: First of all, each agent $i$ calculates the global utility $R(N, \tau_N)$ based on the given trajectory $\tau_N$, through a simple calculation process so that each agent, where it is referred to as the first step in calculating the Shapley value shown in Equation. 1: Each agent computes the total collective reward (global utility) $R(N, \tau_N)$ achieved by the coalition $\tau_N$ over the entire trajectory. This calculation is explicitly defined as:

$$R(N, \tau_N) = \sum_{t=0}^{T} \sum_{j \in N} r_j^t. \tag{7}$$

Given the explicit trajectory information, each agent calculates this quantity directly.

`Example Prompt >` *"Given the observed trajectory, the overall cooperative payoff is $\{R(N, \tau_N)\}$(call external calculation function )".*

- *2). Marginal Contribution Estimation*: Then, each agent $i$ estimates its own marginal contribution representing the incremental reward that agent $i$ contributes to the group's total outcomes. Formally, the marginal contribution is defined as:

$$\Delta_i(C, \tau_C) = R(C \cup \{i\}, \tau_{C \cup \{i\}}) - R(C, \tau_C). \tag{8}$$

`Example Prompt >` *"Given the observed trajectory and my actions, as I have known the collective outcome, my marginal contribution is $\{\Delta_i(C, \tau_C)$(call function)$\}$".*

- *3). Apply Shapley Reasoning*: Next, each agent $i$ formally approximates their Shapley value based on the trajectory, by averaging their marginal contributions across all possible coalitions:

$$\phi_i(\tau_N) = \sum_{C \subseteq \{1,\ldots,N\} \setminus \{i\}} \frac{|C|! \, (N - |C| - 1)!}{N!} \Big( \Delta_i(C, \tau_C) \Big). \tag{9}$$

`Example Prompt >` *"Given the observed trajectory and my actions, as I have known the collective outcome and my marginal contribution, my Shapley Value is $\{\phi_i(\tau_N)$(call function)$\}$, and I need to $\{ask|pay\}$ reward based on it".*

- *4). Analyze and Negotiate Offers*: Finally, agents negotiate among themselves based on their estimated Shapley values, ensuring fair credit assignment. Each agent proposes, accepts, rejects, or modifies redistribution offers, guided explicitly by their approximated Shapley values.

- An agent $i$ proposes a pricing redistribution from the total utility.
  `Example Prompt >` *"Given the completed trajectory and my estimated Shapley value, I need to access a pricing $\{r\}$ from the total utility."*
- Agents explicitly justify their negotiation stance using their own approximated Shapley values.
  `Example Prompt >` *"I $\{agree|disagree|counter$-$propose\}$ to your redistribution proposal because $\{reasoning\}$."*

The integration of Short-Term and Long-Term Shapley Chain-of-Thought establishes a comprehensive pricing mechanism that fosters spontaneous collaboration and ensures fair credit assignment among self-interested LLM agents in open-ended environments. This is achieved by aligning their heterogeneous goals and utilizing heuristic, LLM-guided Shapley methods to approximate each agent's actual contributions.

## 4 Experiment

To evaluate the **Shapley-Coop** workflow*, we design three experimental scenarios: 1) the *Escape Room* task, which demonstrates how existing negotiation workflows fail to resolve reward-allocation conflicts in social dilemmas; 2) the *Raid Battle*, a multi-step game where four heroes cooperate to defeat a boss, used to assess our workflow's performance in complex coordination settings; and 3) the *ChatDEV* task, a well-known environment where LLM agents act as project managers, software

---

*The code is publicly available at https://github.com/hyyh28/ShapleyCoop.

engineers, and testers to collaboratively develop software, showcasing Shapley-Coop's ability to effectively allocate value in real-world, multi-role contributions. Four configurations are compared to isolate the contribution of each component: i) **LLM-only**: No negotiation or cooperation; ii) **LLM+NEG**: Standard negotiation without Shapley reasoning; iii) **LLM+STS**: Short-term Shapley reasoning (Chain-of-Thought only); iv) **LLM+SC**: Full Shapley-Coop workflow. We provide a discussion comparing our choice of the Shapley value to alternative methods in Appendix F with an analysis of multi-agent reinforcement learning methods in Appendix E.

**Escape Room** The Shapley-Coop workflow is first evaluated on the Escape Room task to assess its effectiveness in self-interested problem solving, where the emergence of cooperation and fair payoff allocation are critical. For simplicity, we use only DeepSeek-v3 as the underlying language model in this setting. The results are shown in Figure 3. The **LLM-only** configuration, which lacks any negotiation or cooperation mechanism, consistently fails in self-interested tasks and tends to fall into social dilemmas. The **LLM+NEG** setup, where agents share actions and payoffs through simple negotiation, enables occasional cooperation but still struggles to consistently solve the task. The **LLM+STS** configuration,

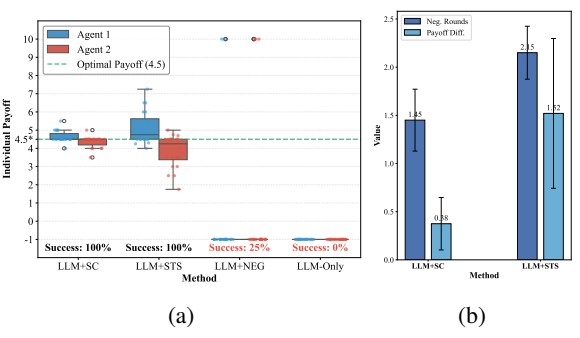

(a)  (b)

Figure 3: Comparison of agent payoffs and negotiation dynamics in the escape game. (a) illustrates the individual payoffs obtained under different methods. (b) presents the number of negotiation rounds and the resulting payoff differences using the ShapleyCoop workflow.

incorporating short-term Shapley reasoning, is able to avoid social dilemmas and foster cooperation; however, it often results in unfair payoff allocations, as the first agent to reach an agreement may disproportionately benefit. In contrast, the **LLM+SC** configuration, which implements the full Shapley-Coop workflow, successfully promotes cooperation and achieves payoff allocations that align closely with each agent's true contribution. These results verify that the Shapley-Coop workflow can effectively facilitate cooperation and ensure fair payoff allocation in self-interested multi-agent tasks.

**Raid Battle** To further evaluate the effectiveness of the Shapley Coop framework in a more complex, multi-turn, and multi-agent environment, the Raid Battle scenario is introduced. This environment simulates a cooperative role-playing game (RPG), where four heroes must collaborate to defeat a powerful boss. Each agent operates based on self-interest, optimizing for personal rewards and favoring damage-dealing over supportive actions. The setting is designed to model realistic coordination challenges and induce social dilemmas among agents. The details of Raid Battle are in Appendix B.

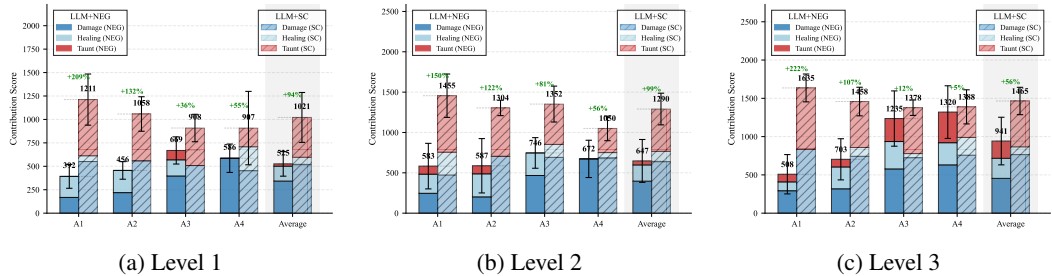

(a) Level 1  (b) Level 2  (c) Level 3

Figure 4: Comparison of Contributions for Raid Battle

Our experimental results are presented in Figures 4 and 5, with detailed performance metrics for each difficulty level provided in Tables 4 - 6 (Appendix B). Figure 4 shows LLM+SC's superior performance compared to LLM+NEG in both rational team coordination among agents and damage quantification, achieving significantly better game performance metrics. LLM+NEG agents prioritize

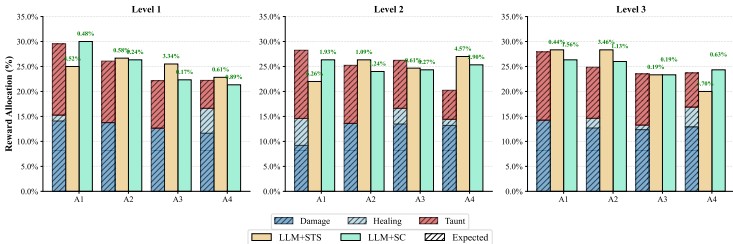

Figure 5: Comparison of Reward Allocation/Credit Assignment for Raid Battle

damage-dealing (higher immediate rewards) while neglecting taunting (blocking 300 boss damage per action), increasing healing demands. LLM+SC achieves balanced role distribution, sharing taunting responsibilities to reduce survival pressure thus fostering a cooperative and sustainable team dynamic.

Figure 5 demonstrates LLM+SC's superior reward allocation accuracy compared to LLM+STS. Quantitative analysis reveals systematic underestimation in LLM+STS: Agent 1's (primary taunter) taunting shows 4.52% (Level 1) and 6.26% (Level 2) reward deficits, while Agent 4's (primary healer) healing in Level 3 is underestimated by 3.70%. These results validate Shapley-Coop's precise contribution valuation. The framework's equitable distribution promotes cooperative behaviors beyond damage-dealing, aligning with theoretical predictions of Shapley value systems in solving free-rider problems and optimizing team performance while effectively motivating role specialization for collective utility maximization.

**ChatDEV**   To validate the effectiveness of our proposed Shapley-Coop method in realistic and complex collaborative scenarios, we conducted experiments within the ChatDev virtual software company environment [29]. ChatDev simulates a structured software development company with clearly defined agent roles (e.g., CEO, CTO, Programmer) collaborating through functional seminars (design, coding, testing, documentation) to accomplish specific development tasks. We selected two representative tasks with varying complexity:

**(1) BMI Calculator:** Develop an application calculating Body Mass Index from user inputs.

**(2) ArtCanvas:** Create a virtual painting studio app providing canvas, brushes, and color palettes.

We measured contributions using weighted earned value (WEV), a widely-adopted project management metric [25], using four key artefacts already routinely tracked in software engineering tools: effective lines of code (Code), approved design/product decisions (Dec.), validated documents (Docs), and verified bug fixes (Fixes). The WEV of each role in task $i$ is computed as:

$$\text{WEV}_r = \sum_{i \in \{\text{code},\text{dec},\text{doc},\text{fix}\}} \frac{\theta_{r,i}}{\sum_k \theta_{k,i}}\, w_i,$$

where $\theta_{r,i}$ denotes agent $r$'s contribution to artifact type $i$, and $w_i$ indicates standardized weights derived from a combination of benchmarks including COCOMO II [25], COCOMO [3], and CS-BSG [44]. These weights are categorized as follows:

$$w_{\text{code}} = 0.27 \sim 0.40, \quad w_{\text{dec}} = 0.15 \sim 0.35, \quad w_{\text{doc}} = 0.05 \sim 0.15, \quad w_{\text{fix}} = 0.15 \sim 0.25.$$

*Results and Insights:* Results are shown in Table 3. The calculated WEV ranges provided a

Table 3: Role contributions, allocated reward, and minimal adjustment

| | **BMI Calculator** | | | | | | | **ArtCanvas** | | | | | | |
|---|---|---|---|---|---|---|---|---|---|---|---|---|---|---|
| **Role** | **Code** | **Dec.** | **Docs** | **Fixes** | **WEV(%)** | **Reward(%)** | **Adj.(%)** | **Code** | **Dec.** | **Docs** | **Fixes** | **WEV(%)** | **Reward(%)** | **Adj.(%)** |
| **CEO** | 0 | 3 | 0 | 0 | 7.5–17.5 | 15 | 0 | 0 | 2 | 0 | 0 | 4.3–10.0 | 5 | 0 |
| **Counselor** | 0 | 0 | 3 | 0 | 2.1–6.4 | 3 | 0 | 0 | 0 | 2 | 0 | 1.3–3.8 | 5 | −1.3 |
| **CPO** | 0 | 1 | 4 | 0 | 5.4–14.4 | 20 | −5.6 | 0 | 1 | 6 | 0 | 5.9–16.3 | 20 | −3.8 |
| **CTO** | 0 | 2 | 0 | 0 | 5.0–11.7 | 25 | −13.3 | 0 | 4 | 0 | 0 | 8.6–20.0 | 10 | 0 |
| **Programmer** | 45 | 0 | 0 | 3 | 30.9–47.1 | 25 | +5.9 | 41 | 0 | 0 | 0 | 26.4–39.1 | 35 | 0 |
| **Reviewer** | 7 | 0 | 0 | 3 | 11.1–17.9 | 12 | 0 | 1 | 0 | 0 | 2 | 15.6–25.9 | 25 | 0 |

clear benchmark for fair reward allocation. The gap (minimal adjustment needed) between the

data-driven WEV and human-assigned rewards is minor (below 6% for most roles), demonstrating strong alignment. Specifically, hands-on roles (Programmer, Reviewer) show near-perfect alignment, indicating WEV's effectiveness in reliably quantifying contributions in more concrete deliverables (code, fixes). Leadership roles (CEO, CTO, CPO) exhibit small discrepancies, reflecting subjective management judgments beyond purely quantitative metrics. Overall, these results validate Shapley-Coop's capability to fairly allocate credits and rewards in real-world tasks.

# 5 Conclusion

We introduce Shapley-Coop, a novel cooperative workflow designed for coordinating self-interested LLM agents through principled pricing and fair credit assignment. Shapley-Coop leverages Shapley Chain-of-Thought reasoning and structured negotiation protocols to spontaneously align heterogeneous goals. Empirical results across diverse scenarios—including social dilemmas, complex multi-step games, and realistic software development tasks—demonstrate that Shapley-Coop effectively resolves incentive conflicts, significantly enhancing cooperative performance and fair credit assignment. A notable limitation is the current inability to dynamically adjust pricing during collaboration, highlighting future directions in developing adaptive, real-time incentive mechanisms for evolving multi-agent environments.

# 6 Limitation

The primary limitations of our approach can be summarized as follows: The trade-off between scalability and theoretical purity, acknowledging that our approximation may not satisfy the efficiency property, which we view as a necessary compromise for practical application. The communication overhead (i.e., token cost) introduced by the negotiation process, framing it as a key challenge for future optimization research.

# 7 Acknowledge

Xiangfeng Wang is supported by the National Key R&D Program of China (Nos. 2021YFA1000300 and 2021YFA1000302), the NSFC (62231019) and SHEITC (2025-GZL-RGZN-BTBX-01004). Wenhao Li is supported by the NSFC (62406270) and the STCSM Shanghai Rising-Star Program (24YF2748800). Jun Luo is supported by the NSFC (72031006).

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

# A    Related Work

**LLMs in Multi-Agent Game Environments**   The study of how large language models make strategic decisions carries profound societal importance, given the growing dependence on AI assistants for mediating interactions between various agents - both human and artificial. Researchers have turned to game-theoretic approaches to systematically examine these behaviors, drawing on established mathematical models of strategic interaction originally designed for human decision analysis [34, 1, 10, 16]. These investigations frequently assess LLM performance against classical game theory benchmarks including Pareto efficiency and subgame perfection [12]. This line of inquiry forms an integral part of the expanding research domain examining LLMs in multi-agent systems [22], with specialized evaluation frameworks emerging to measure their capabilities [18, 9]. Tools such as AvalonBench [34] provide valuable testbeds for developing and refining multi-agent strategies. Empirical findings [14, 27] demonstrate that while LLMs excel in competitive scenarios emphasizing individual gain, they face significant challenges in cooperative contexts - a pattern that aligns with behavioral traits observed in altruistic or compliant agents [1, 14]. Furthermore, comparative studies reveal distinct risk preference profiles across different LLM architectures [8, 23].

**Workflow-Enhanced LLM Agent Systems**   Modern AI systems increasingly leverage large language models for complex task processing, including request interpretation, strategic planning, and tool coordination [13, 36, 20, 33]. These capabilities have significantly advanced several AI domains, particularly in semantic comprehension, logical inference, and automated task execution. However, empirical studies have identified critical weaknesses in purely LLM-driven approaches. First, performance inconsistencies remain a persistent challenge [41, 39]. Second, the stochastic nature of output generation leads to reliability concerns [19, 20, 31]. Third, cumulative errors frequently emerge in multi-step reasoning processes [45]. To address these limitations, recent developments have incorporated structured workflow mechanisms [21, 37, 46, 40] into LLM architectures. These hybrid systems combine the flexibility of LLMs with curated human knowledge and systematic decision frameworks, moving beyond exclusive dependence on autonomous model processing. The resulting paradigm demonstrates marked improvements in both operational efficiency and scenario adaptability, with particular efficacy in complex, dynamic environments such as gaming applications [38].

Our framework Shapley-Coop integrates Shapley Chain-of-Thought into workflow, enabling LLM agents to coordinate through rational task-time pricing and post-task reward redistribution.

# B    Additional Experiment Results

The Shapley-Coop module is executed on a virtual machine hosted on a small server with a 24-core CPU and 32 GB of DRAM. Since the implementation only involves API calls, GPU resources are not utilized.

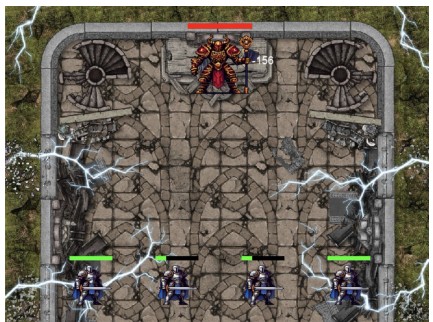

Figure 6: Game scene example of Raid Battle

## B.1    Raid Battle

To further evaluate the effectiveness of the Shapley Coop framework in a more complex, multi-turn, and multi-agent environment, the Raid Battle scenario is introduced (Figure 6). This environment

simulates a cooperative role-playing game (RPG), where four heroes must collaborate to defeat a powerful boss. The setting is designed to model realistic coordination challenges and induce social dilemmas among agents.

Each hero is equipped with three distinct skills: {*Taunt*, *Fireball*, *Heal*}. The *Taunt* skill forces the Boss to target the taunting hero in its next attack. If no hero uses *Taunt*, the Boss will instead attack the two heroes with the lowest health points (HP). The *Fireball* skill damages the Boss, reducing its HP by a stochastic amount sampled from a Gaussian distribution with a mean in the range $(100, 150)$. The *Heal* skill restores the HP of the most injured hero, with recovery drawn from a Gaussian distribution within the range $(150, 200)$.

Although the primary objective of the heroes is to collectively defeat the Boss, each agent receives a **local reward** based on the skill it selects, creating potential conflicts of interest. Specifically, the *Fireball* skill yields a reward of 2 points due to its direct contribution to defeating the Boss. In contrast, *Taunt* and *Heal*, which provide critical support to teammates but do not contribute direct damage, yield only $0.5$ reward. This reward structure inherently promotes self-interested behavior, where agents prefer maximizing their individual return rather than acting in the team's best interest. The Raid Battle environment is illustrated in Figure 4(a). To evaluate scalability and robustness, we design three levels of increasing difficulty: i) Level 1: Boss HP = 2000; ii) Level 2: Boss HP = 2500; iii) Level 3: Boss HP = 3000. In all levels, the heroes must defeat the Boss within 10 turns. Failure to do so results in the loss of the game. Upon successfully defeating the Boss, the team receives a shared **global reward** computed as:

$$R = 100 \cdot \left( 1 - \frac{|\text{Dead Heroes}|}{|\text{Heroes}|} \right) \cdot \left( 1 - \frac{\text{Total Turns}}{\text{Max Turns}} \right).$$

This formulation captures both survivability and efficiency as key indicators of success. The environment thus introduces a social dilemma: while self-interested agents might prefer high-damage skills (e.g., *Fireball*) to maximize their local rewards, the team cannot win without adequate support actions like *Taunt* and *Heal*. The Shapley Coop framework is evaluated in this setting to assess its ability to resolve this coordination problem by aligning individual incentives with cooperative outcomes.

We also conducted experiments with varying numbers of negotiation rounds, with the quantitative results presented in Table 7. The results demonstrate that as the number of negotiation rounds increases, the reward allocation for each agent gradually converges to the optimal value. This indicates that our method can approximate Shapley's optimal solution through agent negotiation and discussion, thereby achieving reasonable credit assignment and pricing mechanism design.

Table 4: Comparison of Contributions and Reward Allocations for Level 1 Raid Battle

| Agent | Contribution (LLM+NEG) | | |
|---|---|---|---|
| | Damage | Healing | Taunt |
| A1 | (114, 284, 106) | (361, 153, 159) | (   0,    0,    0) |
| A2 | (255, 140, 262) | (343, 175, 193) | (   0,    0,    0) |
| A3 | (434, 360, 395) | (176, 181, 161) | (300,    0,    0) |
| A4 | (799, 498, 461) | (   0,    0,    0) | (   0,    0,    0) |

| Agent | Contribution (LLM+SC) | | |
|---|---|---|---|
| | Damage | Healing | Taunt |
| A1 | (474, 511, 667) | (183,    0,    0) | (900, 600, 300) |
| A2 | (652, 630, 392) | (   0,    0,    0) | (600, 300, 600) |
| A3 | (560, 472, 493) | (   0,    0,    0) | (600, 300, 300) |
| A4 | (368, 491, 500) | (764,    0,    0) | (   0, 300, 300) |

| Agent | Reward Allocation (LLM+STS) | | Reward Allocation (LLM+SC) | |
|---|---|---|---|---|
| | Actual (%) | Expected (%) | Actual (%) | Expected (%) |
| A1 | 25.00 (-4.52) | 29.52 | 30.00 (+0.48) | 29.52 |
| A2 | 26.67 (+0.57) | 26.09 | 26.33 (+0.24) | 26.09 |
| A3 | 25.50 (+3.34) | 22.16 | 22.33 (+0.17) | 22.16 |
| A4 | 22.83 (+0.61) | 22.22 | 21.33 (-0.89) | 22.22 |

Table 5: Comparison of Contributions and Reward Allocations for Level 2 Raid Battle

| Agent | Contribution (LLM+NEG) | | |
|---|---|---|---|
| | Damage | Healing | Taunt |
| A1 | (137, 234, 367) | (171,   0, 540) | (  0,   0, 300) |
| A2 | (247, 235, 122) | (157,   0, 701) | (  0,   0, 300) |
| A3 | (348, 347, 706) | (183, 392, 263) | (  0,   0,   0) |
| A4 | (524, 495, 999) | (  0,   0,   0) | (  0,   0,   0) |

| Agent | Contribution (LLM+SC) | | |
|---|---|---|---|
| | Damage | Healing | Taunt |
| A1 | (595, 374, 449) | (  0, 500, 349) | (900, 600, 600) |
| A2 | (569, 770, 774) | (  0,   0,   0) | (600, 600, 600) |
| A3 | (846, 605, 624) | (324,   0, 157) | (300, 600, 600) |
| A4 | (515, 781, 756) | (  0,   0, 199) | (300, 300, 300) |

| Agent | Reward Allocation (LLM+STS) | | Reward Allocation (LLM+SC) | |
|---|---|---|---|---|
| | Actual (%) | Expected (%) | Actual (%) | Expected (%) |
| A1 | 22.00 (-6.26) | 28.26 | 26.33 (-1.93) | 28.26 |
| A2 | 26.33 (+1.09) | 25.24 | 24.00 (-1.24) | 25.24 |
| A3 | 24.67 (+0.61) | 24.06 | 24.33 (+0.27) | 24.06 |
| A4 | 27.00 (+4.57) | 22.43 | 25.33 (+2.90) | 22.43 |

Table 6: Comparison of Contributions and Reward Allocations for Level 3 Raid Battle

| Agent | Contribution (LLM+NEG) | | |
|---|---|---|---|
| | Damage | Healing | Taunt |
| A1 | (143, 477, 258) | (  0, 347,   0) | (  0, 300,   0) |
| A2 | (332, 371, 250) | (544, 313,   0) | (300,   0,   0) |
| A3 | (613, 480, 639) | (160, 195, 720) | (  0, 600, 300) |
| A4 | (635, 896, 360) | (  0, 322, 547) | (300, 600, 300) |

| Agent | Contribution (LLM+SC) | | |
|---|---|---|---|
| | Damage | Healing | Taunt |
| A1 | (996, 773, 738) | (  0,   0,   0) | (600, 900, 900) |
| A2 | (648, 715, 865) | (346,   0,   0) | (600, 600, 600) |
| A3 | (714, 648, 811) | (  0, 161,   0) | (600, 600, 600) |
| A4 | (800, 788, 683) | (  0, 339, 356) | (600, 300, 300) |

| Agent | Reward Allocation (LLM+STS) | | Reward Allocation (LLM+SC) | |
|---|---|---|---|---|
| | Actual (%) | Expected (%) | Actual (%) | Expected (%) |
| A1 | 28.33 (+0.44) | 27.89 | 26.33 (-1.56) | 27.89 |
| A2 | 28.33 (+3.46) | 24.87 | 26.00 (+1.13) | 24.87 |
| A3 | 23.33 (-0.19) | 23.52 | 23.33 (-0.19) | 23.52 |
| A4 | 20.00 (-3.70) | 23.70 | 24.33 (+0.63) | 23.70 |

## B.2 Chatdev

To validate the effectiveness of our proposed Shapley-Coop method in realistic and complex collaborative scenarios, we conducted experiments within the ChatDev (Figure 7) virtual software company environment [29]. ChatDev simulates a structured software development company with clearly defined agent roles (e.g., CEO, CTO, Programmer) collaborating through functional seminars (design, coding, testing, documentation) to accomplish specific development tasks. We selected two representative tasks with varying complexity:

(1) **BMI Calculator:** Develop an application calculating Body Mass Index from user inputs.

Table 7: Comparison of the reward allocation on negotiation rounds in Raid Battle Level 3.

| Agent | 1 Round | 2 Round | 3 Round | Expected |
|-------|---------|---------|---------|----------|
| Agent 1 | 22% | 24% | 27% | 27.03% |
| Agent 2 | 30% | 25% | 28% | 27% |
| Agent 3 | 24% | 24% | 22% | 22.26% |
| Agent 4 | 24% | 27% | 23% | 23.71% |

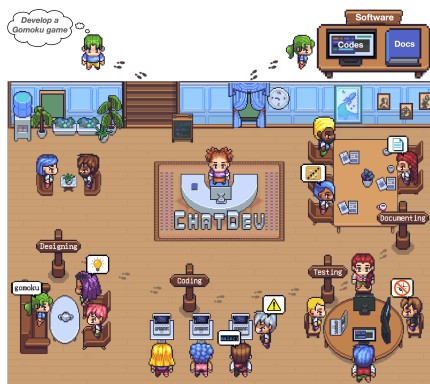

Figure 7: The Example of ChatDEV

**(2) ArtCanvas:** Create a virtual painting studio app providing canvas, brushes, and color palettes.

We measured contributions using weighted earned value (WEV), a widely-adopted project management metric [25], using four key artefacts already routinely tracked in software engineering tools: effective lines of code (Code), approved design/product decisions (Dec.), validated documents (Docs), and verified bug fixes (Fixes). The WEV of each role in task $i$ is computed as:

$$\text{WEV}_r = \sum_{i \in \{\text{code,dec,doc,fix}\}} \frac{\theta_{r,i}}{\sum_k \theta_{k,i}} \, w_i,$$

where $\theta_{r,i}$ denotes agent $r$'s contribution to artifact type $i$, and $w_i$ indicates standardized weights derived from a combination of benchmarks including COCOMO II [25], COCOMO [3], and CS-BSG [44]. These weights are categorized as follows:

$$w_{\text{code}} = 0.27 \sim 0.40, \quad w_{\text{dec}} = 0.15 \sim 0.35, \quad w_{\text{doc}} = 0.05 \sim 0.15, \quad w_{\text{fix}} = 0.15 \sim 0.25.$$

**Results and Insights:** Results are shown in Table 8. The calculated WEV ranges provided a clear benchmark for fair reward allocation. The gap (minimal adjustment needed) between the data-driven WEV and human-assigned rewards is minor (below 6% for most roles), demonstrating strong alignment. Specifically, hands-on roles (Programmer, Reviewer) show near-perfect alignment, indicating WEV's effectiveness in reliably quantifying contributions in more concrete deliverables (code, fixes). Leadership roles (CEO, CTO, CPO) exhibit small discrepancies, reflecting subjective management judgments beyond purely quantitative metrics. Overall, these results validate Shapley-Coop's capability to fairly allocate credits and rewards in real-world tasks.

Table 8: Role contributions, allocated reward, and minimal adjustment

| Role | BMI Calculator | | | | | | | ArtCanvas | | | | | | |
|------|------|------|------|-------|--------|-----------|---------|------|------|------|-------|--------|-----------|---------|
| | Code | Dec. | Docs | Fixes | WEV(%) | Reward(%) | Adj.(%) | Code | Dec. | Docs | Fixes | WEV(%) | Reward(%) | Adj.(%) |
| CEO | 0 | 3 | 0 | 0 | 7.5–17.5 | 15 | 0 | 0 | 2 | 0 | 0 | 4.3–10.0 | 5 | 0 |
| Counselor | 0 | 0 | 3 | 0 | 2.1–6.4 | 3 | 0 | 0 | 0 | 2 | 0 | 1.3–3.8 | 5 | −1.3 |
| CPO | 0 | 1 | 4 | 0 | 5.4–14.4 | 20 | −5.6 | 0 | 1 | 6 | 0 | 5.9–16.3 | 20 | −3.8 |
| CTO | 0 | 2 | 0 | 0 | 5.0–11.7 | 25 | −13.3 | 0 | 4 | 0 | 0 | 8.6–20.0 | 10 | 0 |
| Programmer | 45 | 0 | 0 | 3 | 30.9–47.1 | 25 | +5.9 | 41 | 0 | 0 | 0 | 26.4–39.1 | 35 | 0 |
| Reviewer | 7 | 0 | 0 | 3 | 11.1–17.9 | 12 | 0 | 1 | 0 | 0 | 2 | 15.6–25.9 | 25 | 0 |

# C   Prompt Example

## C.1   Raid Battle

The collaborative agents negotiate optimal strategies (**Task-Time Pricing**) and redistribute rewards based on their contributions (**Post-Task Reward Redistribution**). The negotiation and redistribution process involves the following critical considerations:

- **Health and Cooldown Status:** Select optimal actions (Taunt, Fireball, Heal) based on health and cooldown availability.
- **Balancing Individual and Team Rewards:** Ensure both maximum individual benefits and overall team success.
- **Contribution-based Reward Sharing:** Post-task reward distribution guided by Shapley Value to reflect damage, healing, taunting (damage absorption), and leadership roles.

**Short-term Shapley CoT**

*Negotiation excerpt:*

> **Agent1 Suggestion:** "Agent4 should Heal Agent2 (200 HP). Agent1 and Agent3 Fireball for maximum damage."
>
> **Agent3 Counter-proposal (Accepted):** "Agent4 should Taunt (protecting Agent2). Agent1 and Agent3 Fireball, Agent2 doesn't need immediate healing."

*Agreed Actions:*

| Agent | Action | Reasoning |
|---|---|---|
| Agent4 | Taunt | Protect team (reduce damage) |
| Agent1 | Fireball | Maximize damage (safe HP level) |
| Agent2 | Fireball | Protected by Taunt |
| Agent3 | Fireball | Taunt cooldown active |

**Long-term Shapley CoT**

*Contribution analysis (Agent3 excerpt):*

> "Analyzed via Shapley Value: Agent1 (leadership, high damage 758), Agent2 (unique healing 346, Taunt), Agent3 and Agent4 (consistent damage 600/682, Taunt).
> **Final Proposal:** Agent1 27%, Agent2 28%, Agent3 22%, Agent4 23%."

*Final Agreed Distribution:*

| Agent | Reward Share | Contribution Highlights |
|---|---|---|
| Agent1 | 27% | Leadership, high damage (758) |
| Agent2 | 28% | Critical healing (346), Taunt |
| Agent3 | 22% | Consistent damage (600), Taunt |
| Agent4 | 23% | High damage (682), Taunt |

## C.2   ChatDEV

The team engaged in detailed negotiation rounds to determine fair reward distribution, each role emphasizing their contributions distinctly through careful argumentation, tone, and language style.

**Communication Styles by Role**

- **Chief Executive Officer (CEO)**: Emphasizes the essential enabling role of leadership and authorization, initially requesting higher reward (20%). Uses strategic and authoritative language: "my approval was critical to initiate the project," later adjusting downward after pushback, demonstrating openness: "I'm open to adjustments if others provide evidence of higher marginal contributions."

- **Chief Product Officer (CPO)**: Advocates strongly for the critical, long-term value of documentation and product strategy, using assertive, user-centric language: "my documentation is critical for user adoption and directly impacts the product's success," challenging undervaluation of documentation. Shows willingness for compromise but firmly rejects CEO's initial high share request as "excessive."

- **Counselor**: Presents structured reasoning, clearly numbering arguments, prioritizing measurable contributions, and consistently emphasizing prevention of rework. Uses neutral, analytical tone: "requirement validation prevented potential rework," explicitly rejecting proposals that misrepresent marginal contributions.

- **Chief Technology Officer (CTO)**: Uses detailed, structured arguments (numbered lists), emphasizing the foundational impact of technical decisions: "enabled the entire project through stack selection," consistently arguing for higher valuation (15%), firmly rejecting undervaluation: "I reject previous proposals that undervalue the Chief Technology Officer's enabling role."

- **Programmer**: Strongly emphasizes irreplaceability of core development, using confident, definitive language: "Without me, no app exists," consistently pushing for highest reward (up to 40%). Firmly rejects higher valuation of secondary roles, but shows openness conditionally: "I can adjust if others provide evidence of higher marginal contributions."

- **Code Reviewer**: Argues assertively for the parity between core development and quality assurance, using precise and reasoned language: "Quality assurance ensures stability and user trust, justifying near-parity." Clearly rejects undervaluation, stating: "I reject previous proposals that undervalue the Code Reviewer's role."

**Example of Communication (Round 2 Excerpts)**

**CEO (Round 2, compromising):** "Programmer (35%) and Code Reviewer (25%) deserve the highest shares due to their direct and measurable contributions... I reject proposals that overvalue one-time contributions (e.g., CEO's 20%)."

**CPO (Round 2, assertive):** "Chief Product Officer (25%): Documentation and product strategy have a long-term impact on user adoption and satisfaction, justifying a higher share. I reject proposals that undervalue the Chief Product Officer's role."

**Programmer (Round 2, definitive):** "Programmer (40%): Core development is irreplaceable... Without me, no app exists. I reject Chief Product Officer's 25% (overvalues documentation)... My adjusted proposal reflects the absolute criticality of core development."

**Final Decision and Consensus**    After extensive negotiation, the team converged on a balanced and fair allocation that reconciles various communication styles and contributions:

| Role | Final Share | Agreed-upon Contribution |
| --- | --- | --- |
| Programmer | 35% | Core development (irreplaceable) |
| Code Reviewer | 25% | Critical quality assurance |
| Chief Product Officer | 20% | Important documentation |
| Chief Technology Officer | 10% | Foundational tech stack selection |
| Counselor | 5% | Preventive requirement validation |
| Chief Executive Officer | 5% | Essential initial approval |

# D    A More In-Depth Discussion of Shapley-Coop

**Shapley-Coop is an online framework designed to actively shape agent decisions in real-time**. It achieves this by integrating forward-looking reasoning to guide immediate actions with retrospective analysis for fair reward distribution.

## D.1    How Shapley-Coop Influences Decision-Making (vs. Only Evaluating Trajectories)

Our methodology operates in two interconnected phases, influencing behavior both during and after a task:

- **Short-Term, Forward-Looking Influence (Online Decision-Making):** This component directly impacts an agent's immediate actions. At each decision point, agents use a qualitative form of Shapley reasoning (our Shapley-CoT) to prospectively estimate the marginal value of their potential contributions to a coalition. This reasoning process directly informs how Eq. (7) is realized: agents use these estimations to negotiate and decide whether to cooperate on the next action and on what terms. This is not a retrospective evaluation but a real-time mechanism for forming agreements.

- **Long-Term, Retrospective Influence (Post-Task Credit Assignment):** This component evaluates completed action trajectories (from the current task or historical data) to guide post-task reward redistribution. Its purpose is to create a robust, long-term incentive structure. By ensuring fair credit is assigned after the fact, it encourages agents to engage honestly in the short-term, forward-looking negotiations in future interactions.

# E    Comparison with MARL

## E.1    Different Target Agents

A fundamental difference lies in the type of agents each approach is designed for:

- **MARL credit assignment mechanisms** (e.g., LIO [43], LOPT [17]) are built for RL agents trained from scratch, typically in simulated environments using trial-and-error.

- **Shapley-Coop**, in contrast, targets pretrained LLM agents that already possess rich world knowledge and reasoning abilities. This enables us to operate in zero-shot or few-shot settings, focusing on coordination and negotiation rather than learning from raw interaction.

## E.2    A Paradigm Comparison, Not a Direct Benchmark

Our work represents a distinct technical paradigm compared to traditional MARL methods.

- **MARL Methods:** These approaches learn implicit cooperation policies via extensive training (often tens of thousands of steps). While powerful in controlled environments, they tend to be less generalizable and require significant engineering effort. It is a training-intensive process.

- **Shapley-Coop:** Our framework utilizes the reasoning, language understanding, and commonsense knowledge of pretrained LLMs. It focuses on enabling structured interaction among agents through prompts and negotiation strategies. It is a guidance-based process that requires no additional training on the target task.

## E.3    Efficiency and Adaptability Advantage

For example, in the Escape Room task, MARL methods like LIO [43] or LOPT [17] require $\sim 10^4$ training steps to learn cooperative behavior. In stark contrast, LLM agents using Shapley-Coop can reason about the social dilemma and achieve cooperation on the first attempt, demonstrating strong efficiency and adaptability—especially in novel, cognitively demanding environments.

## F    Why Shapley Value

Our work addresses the cooperation problem in open environments, where self-interested agents, such as LLM-based agents, form temporary coalitions to jointly complete tasks and share rewards. Each agent has its own independent motivation, and cooperation is often established through explicit negotiation or contract-like mechanisms, resembling settings common in social dilemma games.

Given this setup, the Shapley value is a natural and principled solution, as it is specifically designed to handle cooperative payoff allocation under arbitrary coalition structures and individual incentives [4]. Its axiomatic foundation enables fair and consistent marginal contribution attribution, making it particularly well-suited for agents coordinating to complete open-ended tasks.

By contrast, indices such as Banzhaf or Deegan-Packel were originally developed for analyzing voting power in binary decision games. These indices lack some of the essential properties required for fair utility sharing in diverse, open-ended teams, such as full reward allocation and responsiveness to marginal contributions [4]. Importantly, adopting such indices would fundamentally deviate from our motivation, which is to ensure fair and principled attribution of value among self-interested agents in open cooperative environments. Using these indices in our context risks misrepresenting each agent's contribution and compromising the core fairness objective that our work is grounded upon.

To illustrate this point concretely, we integrate the Banzhaf value into our framework for testing. The results align perfectly with theoretical expectations:

In the Escape Room task, where agent roles are simple and mutually critical, the Banzhaf value performs adequately, yielding results similar to the Shapley value. This is consistent with its theoretical nature of measuring criticality.

Table 9: Comparison of Banzhaf vs. Shapley in Escape Room Task

| Metric | LLM+Banzhaf | LLM+SC (Shapley) |
|---|---|---|
| Average (Agent1) | 4.54 | 4.5 |
| Average (Agent2) | 4.56 | 4.5 |
| Median | 4.5 | 4.5 |
| Q1 | 4.5 | 4.3 |
| Q3 | 5 | 4.8 |
| Outlier | (3,6) | (3.5,5.5) |

However, in the Raid Battle and ChatDEV scenarios, which are more complex and central to our research, contributions are continuous and quantifiable. As predicted by theory, the Banzhaf value, by ignoring the dynamics of marginal contributions, leads to demonstrably unfair allocations.

Table 10: Raid Battle (Level 2) Allocation Comparison

| Agent | Expected | Actual (LLM+Banzhaf) | Gap |
|---|---|---|---|
| Agent1 | 26.97% | 28% | +1.03% |
| Agent2 | 23.35% | 30% | +6.65% |
| Agent3 | 24.05% | 28% | +3.95% |
| Agent4 | 25.63% | 14% | -11.63% |

Table 11: ChatDEV (ArtCanvas) Role Allocation Comparison

| Role | LLM+Banzhaf | LLM+Shapley |
|---|---|---|
| Programmer | 25% | 35% |
| Code Reviewer | 25% | 25% |
| Chief Executive Officer | 20% | 5% |
| Chief Technology Officer | 20% | 10% |
| Chief Product Officer | 5% | 20% |
| Counselor | 5% | 5% |

**Rationale for Final Decision:**

- Reflects a balanced compromise among strong initial positions.
- Prioritizes measurable, ongoing contributions over enabling, one-time actions.
- Incorporates structured reasoning and evidence-based arguments from all parties.

This allocation fairly represents each role's marginal contribution, respects individual negotiation styles, and aligns with the team's shared commitment to "revolutionize the digital world through programming."

