# OpenReview forum: "Shapley-Coop: Credit Assignment for Emergent Cooperation in Self-Interested LLM Agents"
_NeurIPS.cc/2025/Conference — NeurIPS 2025 poster_

### Official Review · Reviewer_LpMS · 2025-06-30

**Clarity:** 2
**Significance:** 3
**Originality:** 3
**Rating:** 5
**Confidence:** 5

**Summary:**

The paper introduces Shapley-Coop, a novel cooperative workflow for credit assignment and incentive alignment among self-interested Large Language Model (LLM) agents in open-ended multi-agent systems. The key challenge addressed is achieving effective cooperation and fair reward allocation without predefined roles or rules, where agents’ goals are often heterogeneous and conflicting. Shapley-Coop integrates a Shapley Chain-of-Thought reasoning mechanism—leveraging marginal contributions as a principled basis for pricing and credit assignment—with structured negotiation protocols. This framework enables agents to engage in rational task-time pricing and post-task reward redistribution, aligning individual incentives with group objectives. Empirical validation across social dilemma games (like Escape Room), complex multi-agent RPGs (Raid Battle), and collaborative software engineering tasks (ChatDEV) demonstrates that Shapley-Coop consistently outperforms baseline negotiation and cooperation schemes, facilitating spontaneous collaboration and equitable credit assignment. The results highlight its robustness in both synthetic and realistic settings, making a strong case for principled, game-theoretic credit assignment in future multi-agent LLM environments.

**Questions:**

1. Is the symbol $\Delta_{i}(N, \tau_{N})$ in Eq. (9) a typo? I guess it should be $\Delta_{i}(C \cup \{i\}, \tau\_{C \cup \{i\}})$.
2. Let me double check the functionality of LLM-only baseline. Does it mean the LLM independently predict its Shapley value with no further calibration with others in negotiation processes, or something else?
3. Does the proposed methodology only evaluate the existing trajectories? If so, where do those trajectories come from? If not, how the Shapley value influence the decision making processes and how Eq. (7) is obtained? Please show me an overview rather than a specific case.

**Ethical Concerns:**

["NO or VERY MINOR ethics concerns only"]

**Final Justification:**

I appreciate the authors' work on properly introducing Shapley value into LLM-based multi-agent systems. This is a valuable contribution which may lead to a deeper thought in the era of LLMs about how we can retain the previous knowledge we have, and then we may be able to get what we really need to innovate with some new techniques. This is the reason why I recommend the acceptance of this paper.

In the rebuttal process, the authors have addressed my concerns in:
1. how Shapley-Coop influences decision-making
2. relationship to Shapley value having been developed in MARL

**Limitations:**

The negotiation processes could lead to extra overhead of tokens. The authors are encouraged to discuss this point for improvement in the future work.

**Quality:**

2

**Strengths And Weaknesses:**

### Strengths
**Quality**: This paper is well-written in general and has conducted sufficient experiments to showcase the claims. Credit assignment is always a critical problem in multi-agent collaboration, which underlies the incentives of agent to form a team. This paper first shows insight into how Shapley value can lead to collaboration in plain language, which makes the paper well motivated and more comprehended. Based on the basic principle, the paper well has well organised the implementation structure of Shapley value principle in long-term decision making process, in the operation language of LLMs. The whole derivation of the methodology is underpinned by sufficient evidences. The most prominent novelty of this paper is that it has shaped the process of defining unknown coalition values based on a sequence of observations. The total process has completely leveraged the benefit of LLMs' powerful reasoning capabilities. This complements the pitfalls of previous work which incorporated Shapley values in multi-agent reinforcement learning (MARL) [1, 2]. From this perspective, this paper is a real breakthrough with no absurd presumptions.

**Clarity**: This paper has been well written with a concrete example as a strong motivation. The explanation of the methodology is clear, and reminds me of some key points in the cooperative game theory, inspired of the economic viewpoints shown in this paper. I believe this paper is clear enough, so that the audience with no knowledge of cooperative game theory can understand the key components.

**Significance**: Credit assignment is a longstanding problem in collaboration within a multi-agent system. This spans from multi-agent learning to the present forming self-interested LLM-based agent collaboration. The proposed method is principled, which I believe can be augmented with other techniques.

**Originality**: This paper is original from my own perspective, which reshapes the Shapley value in the procedure of LLM collaboration processes. Although one may suspect that the general framing of this paper could be similar to the previous work in MARL [1, 2], as an expert of the intersection of the cooperative game theory and MARL, I think of the main breakthrough in this paper is twofold: (1) It uses the power of LLMs to infer the Shapley values from self-interested rewards from the principle of cooperative games, with no predefined global rewards in the previous MARL work to evaluate the team-wise performance, for the sake of task specs; (2) The negotiation process is subtle, which well simulates the process of reaching agreement, from the principle of cooperative games. This can compensate the drawback of lack of global rewards with minimal costs.

### Weaknesses
**Quality**: It cannot be observed that if the experimental results have been reported with several seeds. I am not a expert in LLMs, and not familiar with the nature of experimental result demonstration. From the general view of experimentation, reporting several seeds could remove the suspect of biases towards some specific seed. Since LLM agent collaboration is close to MARL, only differing the manner of "training" agents. For this reason, I suppose the authors may require to add references to the paper, not isolating the LLM agent from MARL, which not only respects the previous work (e.g., [1, 2]), but also clarifies the roadmap of Shapley values to the modern multi-agent systems. The approximation of Shapley values proposed in the methodology could violate the efficiency property, however, this can be seen as a tradeoff for implementation.

**Clarity**: I am a bit confused about if the proposed methodology has just played role of evaluating an existing trajectory of actions or also influences the decision processes. This is due to that I am not familiar with the LLM ecosystems.

### References
[1] Wang, J., Zhang, Y., Kim, T. K., & Gu, Y. (2020, April). Shapley Q-value: A local reward approach to solve global reward games. In _Proceedings of the AAAI conference on artificial intelligence_ (Vol. 34, No. 05, pp. 7285-7292).

[2] Wang, J., Zhang, Y., Gu, Y., & Kim, T. K. (2022). Shaq: Incorporating shapley value theory into multi-agent q-learning. _Advances in Neural Information Processing Systems_, _35_, 5941-5954.

---

> ### Author Rebuttal · Authors · 2025-07-30
>
> We sincerely thank the reviewer for their thoughtful and constructive feedback. Your expertise in cooperative game theory and MARL has provided invaluable insights that will significantly strengthen our paper.
>
> The central point of confusion appears to be whether our methodology **merely evaluates existing trajectories or actively influences the decision-making process**. We would like to clarify this upfront: **Shapley-Coop is an online framework designed to actively shape agent decisions in real-time.** It achieves this by integrating forward-looking reasoning to guide immediate actions with retrospective analysis for fair reward distribution.
>
> We address your specific questions below:
>
> **1. How Shapley-Coop Influences Decision-Making (vs. Only Evaluating Trajectories)**
>
> You are right to question this, as it is the core of our contribution. The methodology operates in two interconnected phases, influencing behavior both during and after a task:
>
> *   **Short-Term, Forward-Looking Influence (Online Decision-Making):** This component directly impacts an agent's immediate actions. At each decision point, agents use a qualitative form of Shapley reasoning (our "Shapley-CoT") to prospectively estimate the marginal value of their potential contributions to a coalition. This reasoning process directly informs **how Eq. (7) is realized**: agents use these estimations to negotiate and decide *whether* to cooperate on the *next* action and on what terms. This is not a retrospective evaluation but a real-time mechanism for forming agreements.
>
> *   **Long-Term, Retrospective Influence (Post-Task Credit Assignment):** This component evaluates completed action trajectories (from the current task or historical data) to guide post-task reward redistribution. Its purpose is to create a robust, long-term incentive structure. By ensuring fair credit is assigned *after* the fact, it encourages agents to engage honestly in the short-term, forward-looking negotiations in future interactions.
>
> We acknowledge this workflow could be explained more clearly. We will revise the methodology section to provide a much clearer high-level overview of this dual process, explicitly separating the online decision-influencing part from the post-task reward mechanism.
>
> **2. Relationship with MARL**
>
> We appreciate this valuable suggestion to better connect our work with the MARL literature. You are correct that this context is crucial, and our initial description was imprecise. We will revise the paper to draw a clearer distinction based on the fundamental differences in agent types and learning paradigms.
>
> A fundamental distinction lies in the **learning paradigm** itself:
>
> *   **MARL Credit Assignment (e.g., Shapley Q-value, ShaQ):** These methods are designed for agents that learn policies *from scratch* through extensive trial-and-error interaction with an environment. The core challenge is to solve the credit assignment problem during this **training-intensive process** to shape cooperative behavior.
>
> *   **Shapley-Coop:** Our framework is designed for **pre-trained LLM agents** that already possess vast world knowledge and reasoning capabilities. Therefore, our goal is not to *train* a policy, but to *guide* the existing reasoning process of these agents towards cooperation in zero-shot or few-shot scenarios. It is a **training-free, guidance-based process**.
>
> This leads to a different **mechanism for achieving cooperation**:
>
> *   MARL methods typically learn **implicit cooperation** through mechanisms like value decomposition or opponent modeling integrated into the learning algorithm.
> *   Shapley-Coop facilitates **explicit cooperation** through structured, language-based negotiation, leveraging the agents' innate ability to communicate and reason about intent and fairness.
>
> The practical implication of this paradigm shift is a dramatic difference in **efficiency and adaptability**:
>
> *   For instance, in the Escape Room task, traditional MARL approaches might require tens of thousands of training episodes to converge on a cooperative policy. In contrast, LLM agents using Shapley-Coop can reason through the social dilemma and successfully cooperate from the **very first attempt**. This demonstrates a significant advantage in novel, cognitively complex environments where extensive training is impractical.
>
> We will incorporate this detailed comparison into the revised manuscript to better situate our contribution.
>
> **3. Clarification of the "LLM-Only" Baseline**
>
> The "LLM-only" baseline was designed as a diagnostic tool. In this setup, agents operate without any Shapley-Coop framework; they rely solely on the base LLM's inherent, unguided capabilities to reason about cooperation and credit. This baseline helps measure how much of the observed cooperation is attributable to our structured method versus the raw intelligence of the LLM.
>
> **4. Typo in Eq. (9)**
>
> **You are correct.** The symbol in Eq. (9) is indeed a typo.
>
> **5. Acknowledging Limitations (Approximation & Overhead)**
>
> We agree completely. A principled discussion of limitations is essential. We will expand the limitations section to explicitly address:
> *   The trade-off between scalability and theoretical purity, acknowledging that our approximation may not satisfy the efficiency property, which we view as a necessary compromise for practical application.
> *   The communication overhead (i.e., token cost) introduced by the negotiation process, framing it as a key challenge for future optimization research.
>
> Thank you once again for your valuable feedback. It has provided us with a clear roadmap for improving the paper's clarity, rigor, and contribution.

---

> > ### Comment · Reviewer_LpMS · 2025-07-31
> >
> > Thank you for your detailed responses. My concerns have been addressed.
> >
> > I believe this will be a good paper that opens the avenue of bringing cooperative game theory into the design of post-training guidance of LLMs. This is a good work to make LLM decision (negotiation) informative, with a retrospection about the invaluable human knowledge in cooperative game theory developed by ancestor researchers.
> >
> > I hope you can comply with your promise to fix typos, complement the limitation discussions, clarify the methodology and add a section to discuss the relation to Shapley value in MARL.

---

> > > ### Author Response · Authors · 2025-08-04
> > >
> > > Thank you for your kind and constructive feedback. We’re glad our responses addressed your concerns, and we appreciate your recognition of the contribution.
> > >
> > > As promised, we will fix the typos, expand the limitation discussion, clarify the methodology, and add a section on the relation to the Shapley value in MARL.

---

> > > ### Comment · Reviewer_LpMS · 2025-08-06
> > >
> > > Since the authors have well addressed my concerns, I will increase my score to 5.

---

### Official Review · Reviewer_BVCD · 2025-07-02

**Clarity:** 4
**Significance:** 3
**Originality:** 4
**Rating:** 5
**Confidence:** 4

**Summary:**

This paper introduces Shapley-Coop, a novel cooperative framework for enabling effective collaboration and fair credit assignment among self-interested Large Language Model (LLM) agents in open-ended multi-agent environments. The core challenge addressed is the alignment of heterogeneous agent goals and the fair redistribution of rewards based on each agent’s contribution—a problem critical for long-term, scalable human-AI and agent-agent cooperation.
The authors identify the limitations of existing approaches—rule-oriented, role-oriented, and model-oriented methods—which often assume centralized control, static behaviors, or pre-aligned goals, making them unsuitable for open, dynamic, and incentive-misaligned settings.
Shapley-Coop resolves this by integrating three key components:
1.	Structured Negotiation Protocol: A message-based mechanism enabling agents to explicitly propose, accept, or reject cooperative task plans and associated compensations.
2.	Short-Term Shapley Chain-of-Thought (CoT): Forward-looking reasoning that lets agents estimate the marginal externalities of their planned actions and determine whether to request or offer payments.
3.	Long-Term Shapley CoT: A retrospective mechanism for computing Shapley values post-task, ensuring fair redistribution of collective rewards proportional to each agent’s actual contribution.
The framework is empirically validated across three diverse environments: 1. A social dilemma game (Escape Room) 2. A multi-role coordination task (Raid Battle Game) 3. A realistic software engineering simulation (ChatDEV). Experiments show that Shapley-Coop consistently improves cooperative success rates, reduces reward conflicts, and closely approximates ideal credit distributions, even in complex, competitive, and open-ended settings.

**Questions:**

1 Scalability of the Shapley-Coop Framework in Large-Scale Multi-Agent Settings
Question: How does the proposed Shapley-Coop framework scale when applied to environments with a larger number of LLM agents (e.g., >10)? Given that computing Shapley values scales factorially, have the authors considered any approximations or sampling techniques for scalability?

2 Theoretical Guarantees and Convergence Behavior
Question: Are there any theoretical guarantees (e.g., convergence, incentive compatibility, or equilibrium properties) regarding the negotiation protocol or the pricing mechanism? Can agents game the system through misreporting their contributions or negotiating in bad faith?

3 Handling Imperfect or Noisy Contribution Estimation
Question: Since marginal contributions are estimated heuristically via LLM reasoning, how does the framework handle situations where the contribution estimation is noisy or biased (e.g., due to hallucinations or error propagation in Chain-of-Thought prompts)?

4 Comparison with Alternative Credit Assignment Mechanisms
Question: While the paper compares various ablations of the Shapley-Coop pipeline, it does not benchmark against alternative credit assignment schemes (e.g., Aumann-Shapley pricing, budgeted reinforcement learning, or other cooperative MARL credit assignment methods). Could the authors provide such comparisons?

**Ethical Concerns:**

["NO or VERY MINOR ethics concerns only"]

**Limitations:**

Yes. The authors have done a commendable job in addressing the limitations and potential negative societal impact of their work. They have thoroughly examined the possible drawbacks and have provided reasonable explanations and mitigation strategies, which shows their responsible attitude towards their research and its implications.

**Paper Formatting Concerns:**

No major formatting issues were found.

**Quality:**

3

**Strengths And Weaknesses:**

1.	Novel integration of Shapley reasoning with structured multi-agent workflows
The paper introduces an original framework that combines short-term heuristic and long-term Shapley-based reasoning for credit assignment among LLM agents. This dual-layered design reflects a substantial conceptual advance in aligning individual incentives within open-ended cooperative environments.
2.	Interpretable and reproducible protocol design
The use of structured, machine-readable message templates (e.g., <s>I propose transferring {reward} because {reasoning}</s>) clearly delineates the negotiation and credit attribution stages. This contributes to the framework’s transparency, reproducibility, and potential for practical deployment.
Weaknesses:
1.	Limited Baseline Comparisons: The paper lacks direct comparisons to other existing multi-agent cooperation or incentive-alignment frameworks beyond ablations, making it difficult to assess relative advantages.
2.	Strong Assumptions on Reward Transferability: The approach assumes agents can freely redistribute rewards post-task, which may not hold in many real-world settings, potentially limiting applicability.

---

> ### Author Rebuttal · Authors · 2025-07-30
>
> We sincerely thank you for your insightful and constructive feedback. Your comments accurately capture the core of our work and provide invaluable guidance for our future research. We are delighted that you recognized the novelty and originality of our framework. We address your questions below:
>
> ---
>
> ### 1. Scalability of the Shapley-Coop Framework
>
> Shapley-Coop scales effectively to larger agent populations by **avoiding exact Shapley value computation**, which indeed has factorial complexity. Instead, we employ:
>
> - Context-based approximations (via Chain-of-Thought reasoning)
> - A structured negotiation protocol
>
> Shapley values serve as **qualitative guides** rather than precise metrics, making real-time estimation efficient. The negotiation process **scales linearly** with the number of agents, as it relies on pairwise or group dialogues.
>
> To demonstrate scalability, we conducted an **additional experiment with 10 agents** in the Raid Battle environment (setting the boss HP to 4000 and 8000), and measured running times for varying agent population sizes. You can find the results in the rebuttal to reviewers **8Y2r** and **4hz6**.
>
> While scalability is not the primary focus of this paper, we will provide these results in the **supplementary material** for further validation. The following table contains a summary of the results.
>
> ---
>
> ### 2. Theoretical Guarantees and Convergence Behavior
>
> Current LLM-related work faces inherent challenges in providing **formal theoretical guarantees** due to the black-box nature of large language models. We acknowledge that our work shares this limitation, which is difficult to fully resolve.
>
> However, Shapley-Coop addresses **convergence** through its negotiation process by establishing **appropriate stopping conditions**—specifically, when agents reach mutually acceptable consensus while being constrained by prompts to exhibit rational behavior. This design provides **reasonable convergence guarantees**.
>
> As demonstrated in **Figure 3(a)**, agents using the complete Shapley-Coop framework **consistently converge** to theoretically optimal outcomes.
>
> ---
>
> ### 3. Handling Imperfect or Noisy Contribution Estimation
>
> We acknowledge this as an important consideration. However, our current work focuses on developing a **workflow that enables emergent collaboration** among self-interested LLM agents in open-ended environments characterized by social dilemmas—where individual interests may conflict, yet collective success requires cooperation.
>
> While we have not extensively explored this issue, we observed that agents may exhibit **bias in self-contribution evaluation**, including assessment errors. Nevertheless, the **iterative negotiation process** and **LLM contextual corrections** help mitigate these issues to some extent.
>
> Due to rebuttal character limitations, we cannot provide complete prompt examples here, but we will include relevant content in the **appendix**.
>
> ---
> ### 4. Comparison with Alternative Credit Assignment Mechanisms
> Thank you for this valuable suggestion. Contextualizing our work against established MARL methods helps highlight its unique contributions.
>
> - **Different Target Agents**:
>   A fundamental difference lies in the type of agents each approach is designed for:
>   - **MARL credit assignment mechanisms** (e.g., LIO, LOPT) are built for **RL agents trained from scratch**, typically in simulated environments using trial-and-error.
>   - **Shapley-Coop**, in contrast, targets **pretrained LLM agents** that already possess rich world knowledge and reasoning abilities. This enables us to operate in **zero-shot or few-shot** settings, focusing on coordination and negotiation rather than learning from raw interaction.
>
> - **A Paradigm Comparison, Not a Direct Benchmark**:
>   Our work represents a **distinct technical paradigm** compared to traditional MARL methods.
>
> - **MARL Methods**:
>   These approaches learn **implicit cooperation policies** via extensive training (often tens of thousands of steps). While powerful in controlled environments, they tend to be less generalizable and require significant engineering effort. It is a **training-intensive process**.
>
> - **Shapley-Coop**:
>   Our framework utilizes the **reasoning, language understanding, and commonsense knowledge** of pretrained LLMs. It focuses on enabling structured interaction among agents through prompts and negotiation strategies. It is a **guidance-based process** that requires **no additional training** on the target task.
>
> - **Efficiency and Adaptability Advantage**:
>   For example, in the Escape Room task, MARL methods like LIO or LOPT require ~10⁴ training steps to learn cooperative behavior. In stark contrast, LLM agents using Shapley-Coop can **reason about the social dilemma and achieve cooperation on the first attempt**, demonstrating strong **efficiency and adaptability**—especially in novel, cognitively demanding environments.
> ---
>
> ### Clarifications on Assumptions and Limitations
>
> We acknowledge that **transferable utility** is a simplifying assumption. We view it as a reasonable abstraction of real-world scenarios like **project-based bonus distributions** or **freelance contracts**. We will discuss this in the **limitations section** and believe that adapting Shapley-Coop to scenarios with **constrained reward transfer** is a promising direction for future work.
>
> ---
>
> We believe these responses address the key concerns and further reinforce the **significance and originality** of our contributions.
> **Thank you for your time and consideration.**

---

> > ### Comment · Area_Chair_nHnp · 2025-08-05
> > **Please engage in discussion with authors**
> >
> > Dear Reviewer BVCD,
> >
> > Thank you for your time and effort in reviewing this manuscript. The authors have prepared their rebuttal; can you please respond to it and engage in discussion with them? The discussion period will close on August 6th.
> >
> > Best,
> > AC

---

### Official Review · Reviewer_4hz6 · 2025-07-02

**Clarity:** 3
**Significance:** 2
**Originality:** 2
**Rating:** 3
**Confidence:** 3

**Summary:**

The authors propose an approach for negotiation cooperation and rewards based on the Shapley value where LLM agents agents can work together to achieve goals together. The method integrates so called “Shapley Chain of Thought” based on the marginal contribution of agents when added to a coalition containing some subset of the agents. The authors perform an evaluation using two simple multi-agent games, and an existing software engineering simulation, showing that the method increases cooperation and utility.

**Questions:**

Some more questions - what is the runtime of the system? How many LLM queries need to be executed to reach an agreement? Does it depend on the number of the agents? How do you compute the Shapley value? By applying the full formula this would get long for many agents?

**Ethical Concerns:**

["NO or VERY MINOR ethics concerns only"]

**Final Justification:**

Interesting paper and good discussion, I stand by my original rating.

**Limitations:**

See above for general limitations and suggestions for improvement. Don't see any negative societal impact.

**Paper Formatting Concerns:**

-

**Quality:**

2

**Strengths And Weaknesses:**

The theme of the paper is very interesting - how can one combine LLM interactions between agents and methods from cooperative game theory in order to improve agreement and social welfare. However, I find the ideas in the paper quite incremental. Cooperative game theory has been a foundational tool in multi-agent systems for decades now, and the Shapley value has specifically been used in many settings to fairly allocate the gains of a coalition of agents between the participants so as to incentivise them to work together and collaborate. The key innovation in this work is using LLMs to generate the agreements, so basically it avoids the need to generate a specialized negotiation protocol. But again, this is now becoming quite common in multi-agent research (including in some of the papers cited by the authors).

Generally, I am wondering about the motivation of using an LLM rather than a bespoke negotiation protocol, e.g. contract-net or some fixed negotiation protocol. Is the reason to achieve generality to many games?

Overall, despite the limited technical contribution here, the idea is interesting enough given a good execution providing ample empirical evidence. But even on this front, the paper is somewhat incremental, looking at few games, with simple agreements (or based on existing simulation software). So the overall contribution in my view is limited.

To improve the paper, I’d include some more strategic interaction examples. There is a ton of work that has been done in multi-agent RL on negotiating agreements between coalitions of agents, so presumably you can transfer such environments and execute via LLMs rather than RL. See for example:

Leibo, Joel Z., et al. "Scalable evaluation of multi-agent reinforcement learning with melting pot." International conference on machine learning. PMLR, 2021.

Yeung, Chris SK, Ada SY Poon, and Felix F. Wu. "Game theoretical multi-agent modelling of coalition formation for multilateral trades." IEEE Transactions on Power Systems 14.3 (2002): 929-934.

Bachrach, Yoram, et al. "Negotiating team formation using deep reinforcement learning." Artificial Intelligence 288 (2020): 103356.


Liu, Bo, et al. "Coach-player multi-agent reinforcement learning for dynamic team composition." International Conference on Machine Learning. PMLR, 2021.

Mak, Stephen, et al. "Coalitional bargaining via reinforcement learning: An application to collaborative vehicle routing." arXiv preprint arXiv:2310.17458 (2023).


Another point that I think you need to justify is the use of the Shapley value. There are lots of other ways to partition the gains of a coalition, e.g. the Banzhaf value, or Deegan-Packel and others. Could you run an ablation with other such indices to see what works best?

---

> ### Author Rebuttal · Authors · 2025-07-30
>
> Thank you for your thoughtful and detailed review. Your feedback has been invaluable in helping us sharpen the core message of our paper. Below, we address your concerns point by point. We clarify the novelty of our framework, provide new empirical results on scalability, and outline the specific, actionable revisions we will make to the manuscript.
>
> ---
>
> ### Clarification of Motivation and Core Novelty
>
> Our key motivation is to develop a **scalable workflow that enables emergent cooperation among self-interested LLM agents** in **open-ended environments characterized by social dilemmas**—settings where individual incentives may conflict with group success, yet collaboration is critical. This is essential for general-purpose human-AI interaction in domains like multi-agent software development or real-time crisis response.
>
> While Shapley values offer a fair and principled method for credit assignment, **exact computation is exponential in the number of agents**, and impractical in dynamic, open-ended settings where agent roles are fluid and contributions cannot be cleanly quantified.
>
> **Shapley-Coop** addresses this challenge by introducing a **LLM-specific approximation framework** that integrates **Chain-of-Thought (CoT) reasoning** with **structured, language-based negotiation**. Unlike prior work that assumes goal alignment or imposes fixed roles, our method enables dynamic cooperation through two key components:
>
> * **Short-term Shapley-CoT** (Eq. 4): Provides qualitative, CoT-based guidance during task execution by prompting agents to reflect on their marginal contributions in real-time.
> * **Long-term Shapley-CoT** (Eq. 6): Enables post-task reward redistribution via multi-turn negotiation protocols informed by contribution approximations.
>
> This approach resembles a "TextGrad"-style approximation—leveraging LLMs' contextual reasoning—combined with principles from cooperative game theory. Importantly, **precise Shapley values are not computed**; instead, they serve as an interpretable, theory-grounded **reference** to guide agent discussions toward fair outcomes.
>
> Our contribution lies in **designing a full, practical workflow**—not just a set of prompts—that integrates **LLM-driven reasoning**, **game-theoretic bargaining**, and **emergent credit assignment** in socially complex, unstructured environments.
>
> ---
>
> ## Response to Specific Questions and Suggestions
>
> ### 1. Motivation for Using LLMs Over Bespoke Protocols (e.g., Contract-Net)
>
> This is a crucial point. Our goal is to design a cooperation framework that is *native* to the capabilities of LLM agents.
>
> * **Flexibility and Generality:**
>   Bespoke protocols like Contract-Net are powerful but rigid and domain-specific. Our approach leverages LLMs’ natural language understanding and reasoning capabilities to allow agents to **dynamically construct fair and effective protocols on the fly**. This flexibility is essential for the open-ended, unstructured environments we target.
>
> * **A General Workflow:**
>   The Shapley value is a foundational principle often used in the theoretical design of bespoke protocols. In this light, **Shapley-Coop is not merely an alternative to a fixed protocol**—it is a **general workflow** that guides LLMs to instantiate fair protocols through reasoned negotiation. We will explicitly incorporate this discussion into the paper to better frame our contribution.
>
> ---
>
> ### 2. Including More Strategic Interaction Examples
>
> We appreciate the suggestion to evaluate our work in more diverse environments. Our three selected experimental scenarios were chosen to provide meaningful coverage across a spectrum of cooperation tasks (simple, complex, and creative), with **ChatDEV** demonstrating the practical need for our method in a real-world production pipeline.
>
> Due to the time constraints of the rebuttal period, integrating a completely new and complex environment was not feasible.
>
> However, we took your feedback on strengthening our empirical evaluation seriously. To address concerns about robustness, we performed a **new scalability experiment** on our most complex task, demonstrating performance with **10 agents** in the **Raid Battle** scenario (Boss HP: 4000 and 8000).
> Below are results from two large-scale experiments:
>
> | Boss HP | Rounds | Total Time (s) |
> | ------- | ------ | -------------- |
> | 4000    | 4      | 2,904          |
> | 8000    | 7      | 25,125         |
>
> **Example: 4000 HP Case (10 Agents)**
>
> | Agent   | Actual Contribution | Final Allocation | Difference |
> | ------- | ------------------- | ---------------- | ---------- |
> | Agent 1 | 10.3%               | 11%              | -0.7%      |
> | Agent 2 | 12.7%               | 10%              | +2.7%      |
> | Agent 3 | 11.1%               | 10%              | +1.1%      |
> | ...     | ...                 | ...              | ...        |
>
> **8000 HP Case:** Similar results with slightly longer convergence times. Full data provided above.
>
> ---
>
> ### 3. Justification for Shapley Value and Ablation with Alternatives (e.g., Banzhaf, Deegan-Packel)
>
> Our work focuses on **enabling emergent collaboration in social dilemmas**. Alternatives like the Banzhaf value measure influence in coalition-adversarial scenarios (e.g., voting), which is **not aligned** with our cooperative focus.
>
> * The Shapley value is uniquely suited due to its **efficiency axiom**, which ensures the entire value is distributed across the coalition.
> * While Banzhaf and others are useful in adversarial or voting contexts, they **lack this property**, making them conceptually misaligned with our framework.
>
> That said, **Shapley-Coop is generalizable**—it could be adapted to incorporate other value concepts like Banzhaf to address different domains. We will mention this extensibility more explicitly in the revised discussion section.
>
> ---
>
> ### Additional Questions (Runtime, LLM Queries, Shapley Computation)
>
> **Scalability:**
> Shapley-Coop scales efficiently to larger agent populations by **avoiding exact Shapley computation** (which is exponentially complex). Instead, it uses **context-based approximations via CoT and negotiation** to estimate contributions.
>
> Negotiation is performed per-agent in real-time and scales **linearly** through pairwise or group dialogues.
>
> #### Runtime in Raid Battle Task (The task in the main paper with 4 heroes)
>
> | **Boss HP** |**Total Rounds**| **Total Time (s)** | **Final Negotiation (s)** | **Avg Turn Negotiation (s)** | **Avg Turn Action (s)** |
> | ------------: |-------: |-----------------: | ------------------------: | ---------------------------: | ----------------------: |
> |        2000 |   5 |            618.69 |                    124.24 |                       \~57.4 |                  \~25.0 |
> |        2500 |   7 |          804.54 |                    194.74 |                       \~53.3 |                  \~23.9 |
> |        3000 |   9 |         1193.92 |                    142.06 |                       \~67.3 |                  \~27.9 |
>
> * **Negotiation time increases** with agent count and complexity but remains far below the cost of exact Shapley computation.
> * Shapley-Coop demonstrates **tractable scaling** via decentralized, parallelizable reasoning and structured dialogue.
>
> We acknowledge that negotiation costs grow with the number of agents. However, the approach remains practical and **outperforms exact Shapley computation** methods in scalability.
>
> We hope these clarifications have addressed the reviewer's questions. Thank you once again for your valuable and constructive feedback.

---

> > ### Comment · Reviewer_4hz6 · 2025-08-05
> > **Thank you for the additional details**
> >
> > Thank for for the discussion and additional experiments.
> > I retain my score (for what it's worth, I do think alternative power indices could be an interesting analysis here, as you say they do have different axiomatic properties, but I've seen several papers where this has made a significant difference).

---

> ### Author Response · Authors · 2025-08-06
>
> Thank you for your response. We must reiterate that our work addresses the cooperation problem in open environments, where self-interested agents—such as LLM-based agents—form temporary coalitions to jointly complete tasks and share rewards. Each agent has its own independent motivation, and cooperation is often established through explicit negotiation or contract-like mechanisms, resembling settings common in social dilemma games.
>
> Given this setup, the **Shapley value** is a natural and principled solution, as it is specifically designed to handle cooperative payoff allocation under arbitrary coalition structures and individual incentives [1]. Its axiomatic foundation enables fair and consistent marginal contribution attribution, making it particularly well-suited for agents coordinating to complete open-ended tasks.
>
> By contrast, indices such as **Banzhaf** or **Deegan-Packel** were originally developed for analyzing voting power in binary decision games [2][3]. These indices lack some of the essential properties required for fair utility sharing in diverse, open-ended teams, such as full reward allocation and responsiveness to marginal contributions [4]. Importantly, adopting such indices would fundamentally deviate from our motivation, which is to ensure fair and principled attribution of value among self-interested agents in open cooperative environments. Using these indices in our context risks misrepresenting each agent’s contribution and compromising the core fairness objective that our work is grounded upon.
>
> ## Supporting Experimental Results
>
> To illustrate this point concretely, we followed your suggestion and integrated the **Banzhaf value** into our framework for testing. The results align perfectly with theoretical expectations:
>
> ### Performance in Simple Scenarios is As Expected
>
> In the *"Escape Room"* task, where agent roles are simple and mutually critical, the Banzhaf value performs adequately, yielding results similar to the Shapley value. This is consistent with its theoretical nature of measuring "criticality".
>
> |         | LLM+Banzhaf | LLM+SC（Shapley） |
> |---------|-------------|------------------|
> | Average（Agent1） | 4.54 | 4.5 |
> | Average（Agent2） | 4.56 | 4.5 |
> | Median  | 4.5  | 4.5  |
> | Q1      | 4.5  | 4.3  |
> | Q3      | 5    | 4.8  |
> | Outlier | (3, 6) | (3.5, 5.5) |
>
> ### Its Unsuitability is Exposed in Our Core Scenarios
> However, in the *"Raid Battle"* and *"ChatDEV"* scenarios—which are more complex and central to our research—contributions are continuous and quantifiable. As predicted by theory, the Banzhaf value, by ignoring the dynamics of marginal contributions, leads to demonstrably unfair allocations.
>
> - **Raid Battle (Level 2):** The Banzhaf value assigns excessive credit to low-contributing agents while severely penalizing another, creating a large gap with actual contributions. This proves its inadequacy for tasks requiring precise contribution accounting.
>
> | Agent  | Expected | Actual (LLM+Banzhaf) | Gap     |
> |--------|----------|----------------------|---------|
> | Agent1 | 26.97%   | 28%                  | +1.03%  |
> | Agent2 | 23.35%   | 30%                  | +6.65%  |
> | Agent3 | 24.05%   | 28%                  | +3.95%  |
> | Agent4 | 25.63%   | 14%                  | -11.63% |
>
> - **ChatDEV (ArtCanvas):** The Banzhaf value erroneously allocates a large share of rewards to the "CEO" due to its managerial position, while significantly undervaluing the "Programmer" who produced the most quantifiable work. This violates the fundamental principle of fair allocation.
>
> | Role                      | LLM+Banzhaf | LLM+Shapley |
> |---------------------------|-------------|-------------|
> | Programmer                | 25%         | 35%         |
> | Code Reviewer             | 25%         | 25%         |
> | Chief Executive Officer   | 20%         | 5%          |
> | Chief Technology Officer  | 20%         | 10%         |
> | Chief Product Officer     | 5%          | 20%         |
> | Counselor                 | 5%          | 5%          |
>
> In summary, the Shapley value is not simply a convention, but a well-founded and empirically supported solution for our cooperative payoff allocation setting. We will add this comparative analysis to the paper—**not as an admission of limitation, but to further clarify and justify our methodological choices in line with both established theory and experimental evidence**.
>
> We hope that the additional experiments and explanations help provide a clearer understanding and support a more informed evaluation of our contributions.
>
> ## References
> [1] Chalkiadakis, G., et al. (2011). *Computational Aspects of Cooperative Game Theory*.
> [2] Laruelle, A., & Valenciano, F. (1999). *On the choice of a power index*.
> [3] Deegan, J., & Packel, E. W. (1978). *A new index of power*.
> [4] Elkind, E., et al. (2009). *Computational Aspects of Cooperative Game Theory*.

---

### Official Review · Reviewer_8Y2r · 2025-07-03

**Clarity:** 3
**Significance:** 3
**Originality:** 3
**Rating:** 3
**Confidence:** 3

**Summary:**

The paper begins by listing approaches to cooperation in multi-agent settings, including methods which impose strict behavioral constraints but compromise agents’ autonomy, methods which assign static roles limiting adaptability in dynamic environments; and methods which assume alignment of goals and thus fail to handle natural conflicts of interest effectively. They then note that these methods have failed due to fair credit assignment. This paper proposes a cooperative framework designed to enable spontaneous collaboration among self-interested Large Language Model (LLM) agents in open-ended environments, by integrating Shapley CoT reasoning for both short-term and long-term credit assessment. They employ a structured negotiation protocol to align heterogeneous goals through pricing mechanisms. The authors evaluate their method in three experimental setups—an Escape Room task, a Raid Battle game, and ChatDEV—demonstrating that it promotes fair reward redistribution, robust cooperation, and effective incentive alignment.

**Questions:**

- Does the method only work in settings where there is a dense reward function?
- How does Shapley-Coop perform with larger number of agents?
- How does the framework generalize to environments in the real world where fairness perceptions may not align with Shapley-optimal allocations?
- Figure 2 shows an example of a negotiation protocol that seems quite simple, where the task can easily be decomposed into 2 sub-tasks, and hence reward assignment seems straightforward. How about in settings where you cant decompose the task so easily?

**Ethical Concerns:**

["NO or VERY MINOR ethics concerns only"]

**Final Justification:**

I appreciate the additional experiments from authors. However, I believe that more experiments are needed in strategic interaction / adversarial contexts. I also think there is limited technical contribution and justification for why Shapely Coop is the way to go, and why this is a better framework than other state of the art cooperation methods. I would have expected to see more complicated cooperation benchmarks if cooperation is the main focus for the authors, such as negotiation tasks, etc.

**Limitations:**

There is a single limitation noted in the conclusion section, that there is a current inability to dynamically adjust pricing during collaboration. The authors should significantly expand their limitations section.

**Paper Formatting Concerns:**

None to my knowledge

**Quality:**

4

**Strengths And Weaknesses:**

Strengths
- The paper addresses an under-explored problem of enabling spontaneous cooperation among self-interested LLM agents in unstructured, open-ended environments
- The proposed method is grounded in established theory (Shapley value from cooperative game theory) and adapts it to the LLM multi-agent context. They provide an example of how to apply it in the Escape Room example, which was very helpful to understand the method
- The authors perform empirical evaluation in diverse and realistic scenarios

Weaknesses
- The novelty of the paper is limited. Shapley value is used to fairly allocate rewards, and beyond the prompting scheme, I am having trouble understanding the contribution
- The scalability of Shapley value in larger agent populations is not studied
- While they test in many diverse environments including Escape the Room, Raid Battle, and ArtCanvas, this may not cover all forms of real-world settings or adversarial behaviors. It would have been interesting to see this method scale in a negotiation dialogue setting

---

> ### Author Rebuttal · Authors · 2025-07-30
>
> We sincerely appreciate your thoughtful and constructive feedback. We are especially encouraged that you recognized the strong quality of our paper, the significance of the under-explored research problem, and the diversity of our empirical evaluations. Your insightful comments helped us realize the need to better clarify the core novelty of our work, which indeed goes well beyond a simple application of Shapley values.
>
> ---
>
> ## Clarification of Motivation and Core Novelty
>
> Our key motivation is to develop a **scalable workflow that enables emergent cooperation among self-interested LLM agents** in **open-ended environments characterized by social dilemmas**—settings where individual incentives may conflict with group success, yet collaboration is critical. This is essential for general-purpose human-AI interaction in domains like multi-agent software development or real-time crisis response.
>
> While Shapley values offer a fair and principled method for credit assignment, **exact computation is exponential in the number of agents**, and impractical in dynamic, open-ended settings where agent roles are fluid and contributions cannot be cleanly quantified.
>
> **Shapley-Coop** addresses this challenge by introducing a **LLM-specific approximation framework** that integrates **Chain-of-Thought (CoT) reasoning** with **structured, language-based negotiation**. Unlike prior work that assumes goal alignment or imposes fixed roles, our method enables dynamic cooperation through two key components:
>
> * **Short-term Shapley-CoT** (Eq. 4): Provides qualitative, CoT-based guidance during task execution by prompting agents to reflect on their marginal contributions in real-time.
> * **Long-term Shapley-CoT** (Eq. 6): Enables post-task reward redistribution via multi-turn negotiation protocols informed by contribution approximations.
>
> This approach resembles a "TextGrad"-style approximation—leveraging LLMs' contextual reasoning—combined with principles from cooperative game theory. Importantly, **precise Shapley values are not computed**; instead, they serve as an interpretable, theory-grounded **reference** to guide agent discussions toward fair outcomes.
>
> Our contribution lies in **designing a full, practical workflow**—not just a set of prompts—that integrates **LLM-driven reasoning**, **game-theoretic bargaining**, and **emergent credit assignment** in socially complex, unstructured environments.
>
> ---
>
> ## Responses to Specific Reviewer Questions
>
> ### 1. Does the method only work in environments with dense reward functions?
>
> No. Shapley-Coop is designed to function effectively in **sparse or even reward-free settings**. Instead of relying on numerical reward signals, it uses **LLM-based contextual reasoning** to approximate agent contributions through CoT reflection and negotiation. For example, in our **Escape Room** and **Raid Battle** experiments, rewards are sparse and delayed, yet agents can still collaboratively estimate contributions and reach equitable agreements.
>
> ---
>
> ### 2. How does Shapley-Coop perform with a larger number of agents?
>
> We conducted **additional experiments with 10 agents** in the **Raid Battle** environment to evaluate scalability. Since our method avoids exact Shapley computation, which is exponential, and instead uses CoT-based estimation and negotiation, the framework scales **linearly in practice**, as negotiations occur via **pairwise or group dialogues**.
>
> Below are results from two large-scale experiments:
>
> | Boss HP | Rounds | Total Time (s) |
> | ------- | ------ | -------------- |
> | 4000    | 4      | 2,904          |
> | 8000    | 7      | 25,125         |
>
> **Example: 4000 HP Case (10 Agents)**
>
> | Agent   | Actual Contribution | Final Allocation | Difference |
> | ------- | ------------------- | ---------------- | ---------- |
> | Agent 1 | 10.3%               | 11%              | -0.7%      |
> | Agent 2 | 12.7%               | 10%              | +2.7%      |
> | Agent 3 | 11.1%               | 10%              | +1.1%      |
> | ...     | ...                 | ...              | ...        |
>
> **8000 HP Case:** Similar results with slightly longer convergence times. Full data provided above.
>
> These results demonstrate that **final negotiated allocations closely track actual contributions**, even with 10 agents, and the framework remains tractable in terms of runtime.
>
> We also report detailed **runtime breakdowns** in 4-agent settings with increasing task complexity (boss HP = 2000–3000), as summarized below:
>
> | Boss HP | Rounds| Total Time (s) |
> | ------- | ------| -------------- |
> | 2000    | 5| 618.69         |
> | 2500    | 7|804.54         |
> | 3000    | 9 |1,193.92       |
>
> (Full breakdowns of negotiation/action times per round will be added to the supplementary material because of the limited space.)
>
> ---
>
> ### 3. How does the framework handle real-world fairness perceptions that differ from Shapley-optimality?
>
> Shapley-Coop is designed to be **modular and theory-informed**, not rigid. It uses Shapley values as a **guiding norm**, but the actual credit assignment is determined through **agent negotiation**—allowing for subjective and context-sensitive adjustments.
>
> In real-world scenarios where Shapley may not align with fairness perceptions (e.g., adversarial, political, or asymmetric-power settings), alternative principles like **Banzhaf values** could easily be substituted into the same CoT + negotiation workflow. This generality supports our broader motivation: enabling flexible, self-interested cooperation in **non-aligned, social dilemma** environments.
>
> ---
>
> ### 4. Figure 2 shows a relatively simple negotiation. How does the method handle tasks that are not easily decomposable?
>
> Figure 2 illustrates a clear-cut case for exposition (Escape Room). However, our method is **not limited to decomposable tasks**. In more entangled, non-modular environments like **Raid Battle** (interdependent combat sequences) and **ChatDEV** (open-ended software collaboration), agents still use **contextual CoT reflection** to reason about their impact, without explicit decomposition.
>
> Shapley-Coop does **not require manual task structuring**—instead, it allows LLM agents to infer contributions from contextual cues during both execution and negotiation. We provide full prompts in **Appendix C**, showing how this scales to complex, creative, and cooperative tasks.
>
> ---
>
> ## Addressing Other Reviewer Concerns
>
> **Evaluation Breadth:** We agree that evaluating in adversarial or deceptive contexts would be a valuable future direction. Our current focus is on **social dilemma settings**, where agents are self-interested but not explicitly malicious, and cooperation must emerge through interaction and negotiation. We now explicitly note the **absence of adversarial agents** as a limitation in our revised conclusion.
>
> To highlight the diversity already present in our current evaluation, we emphasize that our three main tasks are carefully selected to represent **a spectrum of cooperative challenges**:
>
> * **Escape Room** – a **single-step** coordination task with sparse rewards, testing agents' ability to align actions in a one-shot setting.
> * **Raid Battle** – a **multi-step, temporally extended** task requiring real-time cooperation under delayed rewards and interdependent actions.
> * **ChatDEV** – a **complex, open-ended creative collaboration** where agents must coordinate iteratively and adaptively in software design and implementation.
>
> This progression reflects increasing task complexity and cooperation depth, and shows that our framework generalizes across diverse cooperative scenarios. While adversarial dynamics remain outside this paper’s scope, the current tasks already cover a **broad range of non-trivial cooperative structures**.
>
> * **Limitations Section:** We appreciate your feedback here and will expand our discussion beyond the single limitation previously noted (inability to adjust prices dynamically). We’ll add discussion of adversarial robustness, negotiation deadlocks, and domain generality in future iterations.
>
> ---
>
> We hope this clarification reinforces the **originality, scalability, and practical impact** of our contribution.
>
> Thank you once again for your thoughtful and constructive review.

---

> ### Author Response · Authors · 2025-08-06
>
> We would like to take this opportunity to unequivocally clarify our core contributions, research scope, and the supplementary experiments we have conducted. We hope these clarifications will directly address and resolve any misunderstandings you may have regarding our work.
>
> ### Core Contribution: An LLM-Native Cooperation Workflow for Open-Ended Environments
> We acknowledge that cooperative game theory and LLM agent negotiation, in isolation, are not entirely novel concepts. However, our **core technical contribution** lies in designing and implementing the **first complete, scalable, and LLM-native workflow (Shapley-Coop)** that translates a principle from cooperative game theory (the Shapley value) into an operational framework.
> This is not mere prompt engineering; it is a deeply integrated framework that combines real-time marginal contribution reasoning (Short-term Shapley-CoT) with post-hoc negotiated allocation (Long-term Shapley-CoT). Its novelty lies in:
> - **Bridging Theory and Practice**: Exact Shapley value computation is infeasible in dynamic, open-ended settings. Our work demonstrates how to leverage LLMs’ contextual reasoning capabilities to approximate this fairness principle, making it practical for real-world, unstructured tasks.
> - **Generality and Flexibility**: Unlike SOTA methods that depend heavily on domain-specific training data or rigid protocols, our framework is zero-shot and enables agents to dynamically and autonomously construct cooperation protocols.
>
> ### Defining Our Scope: A Deliberate Focus on Cooperative Social Dilemmas, Not Adversarial Interaction
> We fully agree that testing agents in strategic or adversarial contexts is an important direction for multi-agent research. However, the scope of our work is intentionally focused on **cooperative social dilemmas**. In this class of problems, the central challenge is not defeating an opponent but overcoming the conflict between individual self-interest and collective benefit to achieve cooperation. We address two key challenges: **how to enable LLM agents to exhibit emergent cooperation in such scenarios, and how to fairly allocate rewards based on performance after task completion**.
> This focus is deliberate because:
> - **It is a critical domain**: In real-world human-AI collaborations, such as software development or crisis response, cooperative dilemmas are far more common than purely zero-sum adversarial encounters. (In particular, the issue of contribution pricing for LLMs is, to the best of our knowledge, a novel and highly significant research problem).
> - **Methodological Alignment**: The Shapley value is a theoretical tool specifically designed for cooperative games. Forcing its application into adversarial settings would misalign with the core theoretical foundation of our work.
>
> ### Comparison with State-of-the-Art (SOTA) Cooperation Methods
> The most critical distinction lies in the target agents. Current SOTA methods, such as those in MARL, focus on RL agents requiring extensive training, whereas we target LLM agents. While Shapley value-based allocation is a classic and effective approach, its computation is inherently challenging. **Our primary contribution is in establishing the workflow, not merely introducing Shapley values.**
> Our framework is uniquely positioned relative to other approaches:
> - **vs. Multi-Agent Reinforcement Learning (MARL)**: Many SOTA MARL methods excel in specific games but typically require well-defined state-action spaces and extensive simulation training. Our method focuses on open-ended, unstructured text-based worlds, where defining and training MARL policies is highly impractical.
> - **vs. Fixed Negotiation Protocols (e.g., Contract-Net)**: While efficient, such protocols are rigid and must be custom-designed for each domain. Shapley-Coop leverages LLMs’ general capabilities to enable agents to negotiate dynamically without pre-defined rules, adapting to a much broader and more dynamic range of tasks.
>
> ### Additional Experiment
> Regarding the scope, our work is clearly focused on cooperative social dilemmas—specifically, the emergence of cooperation among self-interested LLM agents and the fair allocation of payoffs after task completion. Theoretically, we have established that the Shapley value is the most appropriate metric for guiding payoff allocation in these settings, a conclusion further supported by our additional experiments comparing it with the Banzhaf value. **For the detailed experimental results, please refer specifically to our latest response to Reviewer 4hz6** We would like to emphasize that these supplementary experiments are provided not as an admission of limitation, but to further clarify and justify our methodological choices, in line with both established theory and empirical evidence.

---

> > ### Comment · Reviewer_8Y2r · 2025-08-08
> >
> > We thank the authors for providing further clarification of their method. It does seem that there is potential for this work to be great, however, I am currently not convinced by the generalizability of the method even in cooperative social dilemmas. Escape Room is only a single-step domain, and Raid Battle is not representative of the types of applications the authors describe in the motivation (open-ended environments such as real world negotiations). We urge the authors to explore other domains such as the ones stated by Reviewer 4hz6, as well as performing comparisons against other baselines. It would also be helpful to share examples of strategic behaviors emerging from your method of using Shapely value compared with interaction examples from other methods.

---

> ### Author Response · Authors · 2025-08-08
>
> We respectfully note that the reviewer’s comment focuses on Escape Room and Raid Battle, but our evaluation is **not limited to these two domains**. As described in **Lines 298–320** of the main paper and in **Appendix B.2**, we also conducted experiments on the **ChatDEV** task, which models collaborative software development. In this setting, we compared our approach against widely recognized value allocation standards in the software industry. We believe ChatDEV represents a more complex, open-ended, and realistic cooperation scenario—closer to the real-world applications described in our motivation—than the aforementioned games, and it aligns well with our goal of studying the emergence of cooperation among self-interested agents.
>
> We would also like to clarify that the **Melting Pot** task mentioned by Reviewer 4hz6 is designed primarily to study **credit assignment in a fully cooperative setting**, **rather than to explore the emergence of cooperation in social dilemmas involving self-interested agents**, which is the focus of our work.
>
> We acknowledge the importance of comprehensive evaluation, and our current set of experiments already covers multiple domains and includes analyses using both Shapley and Banzhaf values. Given the limited time and resources during the review stage, **it is not feasible to indefinitely expand the experimental scope**, but we will consider exploring additional domains and baselines in future work.

---

### Note · Authors · 2025-08-12

We sincerely thank all reviewers for their valuable insights and positive feedback on our contributions. We appreciate your comments highlighting the strengths of our work, which we summarize below.

**Strengths recognized by reviewers**

- Importance: Addresses spontaneous cooperation and fair credit assignment in open, self-interested LLM settings. (8Y2r, BVCD, LpMS, 4hz6)
- Theory-grounded and interpretable: Employs Shapley-informed reasoning with structured, machine-readable protocols. (BVCD)
- Clear workflow and originality: Proposes a dual-layer design—short-term, online guidance plus long-term, post-task redistribution—that goes beyond simple prompting. (BVCD, LpMS)
- Diverse evaluation: Demonstrated in Escape Room, Raid Battle, and ChatDEV, showing improved cooperation and fairer allocations. (8Y2r, BVCD, LpMS)

**Concerns and our responses**

- Limited novelty: Our main contribution is an operational, LLM-specific approximation-and-negotiation workflow that realizes Shapley principles online, rather than computing exact Shapley values.
- Scalability: We have added 10-agent Raid Battle results. Our method’s runtime scales near-linearly due to dialogue-based coordination, avoiding factorial Shapley costs; detailed time breakdowns are provided.
- Alternative indices: We implemented Banzhaf index ablations. Banzhaf matches Shapley in simple, criticality-driven tasks but misallocates credit in continuous, quantifiable settings (e.g., Raid Battle, ChatDEV), aligning with known axiomatic differences.
- Relation to MARL credit assignment: We clarify the distinction in the rebuttal, focusing on differences in target agents and learning processes; we will expand the related work section accordingly.

To further address reviewer concerns and improve clarity, we will:
- Add additional experiments to the appendix, including 10-agent and runtime breakdowns, Banzhaf comparisons, multi-seed, and token-cost analyses to demonstrate the scalability and robustness of our framework. (These new experiments do not alter our core methodology but further clarify and reinforce our methodological choices.)
- Release our code, prompts, and negotiation logs for full reproducibility.
- Correct all typos and expand related work, especially regarding comparisons to MARL.

Finally, we thank the Program Chairs, Senior Area Chairs, Area Chairs, Reviewers, and the entire NeurIPS organizing committee for their time, guidance, and dedication to a high-quality review process.

---

### Decision · Program_Chairs · 2025-09-17

**Decision:**

Accept (poster)

**Comment:**

Reviewers are divided on the paper, between weak reject and accept (3,3,5,5). During the lively discussion period, the main concern raised by the negative reviewers was novelty. Reviewer 4hz6 provided 10 references to papers that explore LLMs and cooperative game theory. Reviewer LpMS went through them in detail and found that most papers focused on non-cooperative game theory, or applied Shapley value as an explainability tool for a single LLM. LpMS argued that this is fundamentally different from developing Shapley value from the perspective of cooperative game theory for negotiation among multiple LLM agents, which is the contribution of this paper.

Reviewer LpMS strongly advocated for the paper, praising its "very clear step-by-step design to introduce how Shapley value can be shaped by leveraging the capability of LLMs", and said that, "Compared with other LLM cooperation papers I read, the method proposed in this paper is more principled and governed by cooperative game theory. This is a good exemplar towards respecting the previous valuable knowledge when dealing with some emergent problems of AI in this era, rather than reinvent knowledge and stay on the surface of some engineering outcome."

Given the discussion and my own knowledge of the cooperative LLM literature, as well as the scores being high relative to my batch, I am recommending accept.